

# Improving forecasts of snow water equivalent with hybrid machine learning

Oriol Pomarol Moya[1], Madlene Nussbaum[1], Siamak Mehrkanoon[2], Philip D. A. Kraaijenbrink[1], Isabelle Gouttevin[3], Derek Karssenberg[1], and Walter W. Immerzeel[1]

[1]Department of Physical Geography, Faculty of Geosciences, Utrecht University, Utrecht, The Netherlands
[2]Department of Information and Computing Sciences, Faculty of Science, Utrecht University, Utrecht, The Netherlands
[3]Univ. Grenoble Alpes, Université de Toulouse, Météo-France, CNRS, CNRM, Centre d'Études de la Neige, Grenoble, France

**Correspondence:** Oriol Pomarol Moya (o.pomarolmoya@uu.nl)

**Abstract.** Accurate characterization of snow water equivalent (SWE) is important for water resource management in large parts of the Northern Hemisphere, but its large spatio-temporal variability and limited observational data make it difficult to quantify. Complex physically-based models have been developed that allow long-term SWE prediction, including scenarios without snowpack observations or in future events. However, those still suffer from large errors in their simulations, have long

run times at large scales and provide challenges for integrating observational data. There have been attempts at using machine learning (ML) to improve SWE forecasting from meteorological data with promising results, but the data scarcity issue and concerns about the ability to extrapolate in time and space remain. In this study, we evaluate two hybrid setups that integrate physically-based simulations and ML. The first setup, referred to as post-processing, follows a common approach in which the simulated outputs from a numerical snow model, Crocus, are used as predictors to the ML component in addition to the

meteorological data. The second setup, named data augmentation, involves an ML model trained not only on measured SWE but also on Crocus-simulated SWE at additional locations. These approaches are deployed using *in-situ* meteorological and SWE measurements available at ten stations throughout the Northern Hemisphere, and compared to Crocus and a ML setup using measured data only. The results show that the post processing setup outperforms all other approaches when predicting on left-out years in the training stations, but performs poorly when extrapolating to other locations compared to Crocus. The

addition of a large set of Crocus-simulated variables besides SWE in the post-processing setup results in similar performance for left-out years but exacerbates the spatial extrapolation issue. On the other hand, the data-augmentation setup performs slightly worse on the left-out years, but showed much better transferability to new locations, improving the other ML-based setups greatly and reducing the RMSE in Crocus by more than 10%. The feature importances of the ML-models are consistent with physical knowledge, despite having unusual deviations at extreme values, which could be further improved with the data-

augmentation setup. Lastly, the addition of lagged variables results in improved results, but they are only relevant for up to a week. These results prove the usefulness of hybrid models and particularly the data-augmentation setup for SWE prediction even in data-scarce domains, which has the potential to improve forecasts of SWE at unprecedented spatio-temporal scales.





# 1 Introduction

The cryosphere has a large impact on the Northern Hemisphere, influencing landscapes, ecosystems, and water cycles. Snow, in
particular, acts as a natural water reservoir, regulating seasonal runoff that impacts human socioeconomic activities both locally
and downstream (Beniston et al., 2018; Biemans et al., 2019). Therefore, reliable estimates of snow water equivalent (SWE)
are essential for accurate water resources assessment at various temporal and geographical scales. Nonetheless, important chal-
lenges remain for its quantification due to its spatio-temporal variability. Considerable attention has been given to developing
detailed land-surface and snow models for SWE and hydrological applications (e.g., Tarboton and Luce, 1996; Marks et al.,
1999). Such models have also been used for avalanche forecasting, as is the case with Crocus, a complex physically-based
snow model (Vionnet et al., 2012; Brun et al., 1989). When forced with reanalysis meteorological data, it has shown similar
or better performance compared to other SWE products (Brun et al., 2013; Mortimer et al., 2020), but still exhibits significant
discrepancies when compared to observed or field-derived snow conditions (Lafaysse et al., 2017; Menard et al., 2021).

During the last decades there has been a rapid increase in the application of machine learning models in hydrology, as they
can improve the performance of traditional data-driven and physically-based modelling approaches thanks to their ability to
automatically find non-linear structure in observed data and can easily adapt to multiple scales (Mosaffa et al., 2022). However,
the success of such methods often depends on the availability of large, high-quality, standardized datasets, which are lacking
in the case of SWE. Most previous attempts to estimate SWE with ML have relied on *in situ* snow depth measurements (Odry
et al., 2020; Khosravi et al., 2023; Ntokas et al., 2021) or remote sensing data (Tedesco et al., 2004; Bair et al., 2018; Guo et al.,
2003; Moradizadeh et al., 2023; Santi et al., 2022; Zheng et al., 2018) as inputs. Hence, they cannot be used for forecasting
and are not suitable for prediction over long-time periods without snow data, as can occur at sites with no snow measurements
or only over a limited time-period in the past. Nevertheless, recent studies have shown promising results when predicting daily
SWE using only meteorological and static features (Duan et al., 2024; Wang et al., 2022).

A recent trend in scientific applications of ML is the combination with physically-based models (Reichstein et al., 2019).
The resulting hybrid models rely less on measured data and conform better to known physical laws, which can improve their
predictions compared to purely data-driven approaches, especially in data-scarce scenarios. Due to its easy implementation,
one of the most common hybrid approaches is to use the output of a physically-based model as additional features to the
ML model (Willard et al., 2022, Section 3.4.2). This "post-processing" approach has also been applied in the context of snow
modelling. For instance, King et al. (2020) used a random forest to correct biases from a modelling and data assimilation SWE
product in Ontario, and Steele et al. (2024) integrated outputs from a physically-based model into a hybrid LSTM framework
to predict SWE and snow depth in several stations across the western United States. However, no studies have comprehensively
evaluated both temporal and spatial extrapolation capabilities of these type of models across diverse geographic regions for the
purpose of SWE prediction. Alternatively, hybrid models that augment the training data using synthetic data simulated with
physically-based models show promise for predicting in data-scarce scenarios. Nevertheless, while this modelling technique
has obtained good results for other hydrological applications such as streamflow prediction (López-Chacón et al., 2023), its
applicability for snow forecasting remains unknown.





The primary objective of this study is to find a method that can predict point-observations of SWE with high precision at daily resolution over long time periods, i.e., from months to several years or decades. To do so, we evaluate the performance of the two previously introduced hybrid models, namely the post-processing (PPC) and data-augmentation (AUG) approaches, for predicting the change in SWE given a set of meteorological features. *In situ* snow and meteorological data from ten stations across the Northern Hemisphere are used together with snowpack simulations at the same locations using the Crocus snow model. The models are evaluated for 1) forecasting in locations for which historical SWE measurements are available and 2) targetting SWE prediction at ungauged stations. The SWE simulations from Crocus and the outputs of a fully measurement-based ML model (MSB) are employed as a comparative benchmark. Furthermore, several aspects of model creation are explored, such as testing for a range of ML algorithms and hyperparameters and incorporation of lagged or additional modelled state variables as inputs. Finally, the importance of the input features is explored to investigate the physical plausibility of the ML predictions.

## 2 Data and methods

### 2.1 Measured SWE and meteorological data

The meteorological and SWE data used in this study correspond to the ESM-SnowMIP meteorological and evaluation datasets, an international project that aimed to assess and compare snow modelling schemes (Krinner et al., 2018). The data was collected from ten stations throughout the northern hemisphere including seven to twenty years of *in situ* measurements. A full description of these stations can be found in Menard and Essery (2019). The meteorological data was reported at hourly resolution and include the surface atmospheric pressure (Pa), the near-surface specific humidity (kg kg$^{-1}$), air temperature (K) and wind speed (m s$^{-1}$), the rainfall (kg m$^{-2}$ s$^{-1}$) and snowfall (kg m$^{-2}$ s$^{-1}$) rates, and the surface downward longwave (W m$^{-2}$) and shortwave (W m$^{-2}$) radiations. Snow water equivalent (mm) was reported at varying time intervals depending on the station, from hourly to bi-weekly. For three of the stations automatic measurements from a snow pillow or cosmic ray sensor were provided at daily resolution, but for the remaining stations only manual observations at irregular intervals were available. These will be referred to as automatic and manual stations, respectively.

The limiting factor in terms of temporal resolution was the SWE measurements, so all data was resampled from hourly to daily frequency. For the measured SWE data, the value at 12:00 was selected, as it corresponds to the hour for which most measurements were taken. Regarding the meteorological data, the 24 hours in-between SWE measurements were aggregated by computing their average (avg). Additional aggregation methods were performed for some variables according to expert knowledge, specifically: the maximum value (max) of the rainfall rate and wind speed, the time integral (int) of the positive air temperatures in Celsius degrees, and the average during daytime (dav) of the specific humidity and both shortwave and longwave radiations, according to the geographical location of each station. The surface pressure was not used as a predictor since it was lacking measured data for some stations. The final aggregated variables fed to the ML models as input are described in Table 1.



**Table 1.** Description of daily-aggregated meteorological variables used as input for the machine learning models.

| Variable | Description |
| --- | --- |
| Qair_avg | Average of the near-surface specific humidity ($\mathrm{kg\,kg^{-1}}$) |
| Qair_dav | Daytime averaged near-surface specific humidity ($\mathrm{kg\,kg^{-1}}$) |
| Rainf_avg | Average of the rainfall rate ($\mathrm{kg\,m^{-2}\,s^{-1}}$) |
| Rainf_max | Maximum rainfall rate ($\mathrm{kg\,m^{-2}\,s^{-1}}$) |
| Snowf_avg | Average of the snowfall rate ($\mathrm{kg\,m^{-2}\,s^{-1}}$) |
| LWdown_avg | Average of surface downward longwave radiation ($\mathrm{J\,m^{-2}}$) |
| LWdown_dav | Daytime averaged surface downward longwave radiation ($\mathrm{J\,m^{-2}}$) |
| SWdown_avg | Average of surface downward shortwave radiation ($\mathrm{J\,m^{-2}}$) |
| SWdown_dav | Daytime averaged surface downward shortwave radiation ($\mathrm{J\,m^{-2}}$) |
| Tair_avg | Average of the near-surface air temperature (°C) |
| Tair_int | Positive integral of the near-surface air temperature (°C) |
| Wind_avg | Average of the near-surface wind speed ($\mathrm{m\,s^{-1}}$) |
| Wind_max | Maximum near-surface wind speed ($\mathrm{m\,s^{-1}}$) |

## 2.2 Crocus snowpack simulations

SWE and other snowpack variables coming from Crocus model simulations, generated for the ESM-SnowMIP project, were used in this study as part of the hybrid modelling approaches and as a benchmark for model evaluation. Crocus considers the energy and mass balance of the snowpack to model its evolution with high physical detail. It dynamically adjusts up to 50 layers to represent snow stratigraphy and provides a comprehensive evolution of the snow microstructure. Crocus was forced with the aforementioned meteorological data to simulate a one-dimensional snowpack column at the ten ESM-SnowMIP stations with

hourly resolution. The model was run without calibration and was coupled to the soil component of the land surface scheme ISBA (Vionnet et al., 2012), which tracks the temperature and moisture of 20 soil layers.

To conform to the daily frequency of the measured data, the Crocus-predicted SWE at 12:00 was selected for each date. Besides SWE, Crocus reports a range of bulk snowpack and snow layer variables that could potentially be used in the PPC setup. The layer information was not directly included, but was used to generate two additional bulk variables not reported in

Crocus: the cold content, calculated as the sum of the layer-wise product of SWE, snow temperature, and specific heat of ice, set to $2100\,\mathrm{J\,kg^{-1}\,K^{-1}}$ (Jennings et al., 2018); and the snow bulk saturation, computed as the sum of the snow liquid content for all layers divided by the depth of the snowpack. All Crocus snowpack variables were resampled to daily frequency following the same procedure as the meteorological variables. Besides the averages, only the maximum daily surface temperature was added. However, for the snow depth and cold content, the value at the current time step (vcs) was considered more relevant





**Table 2.** Description of daily-aggregated model state variables used as input for the machine learning models.

| Variable | Description |
| --- | --- |
| Soil_temp_layer_0_avg | Average of the temperature in the top soil layer (K) |
| Soil_liquid_layer_0_avg | Average of the relative amount of liquid water in the top soil layer ($\mathrm{m^3\,m^{-3}}$) |
| Soild_ice_layer_0_avg | Average of the relative amount of ice in the top soil layer ($\mathrm{m^3\,m^{-3}}$) |
| RN_ISBA_avg | Average of the net radiation ($\mathrm{W\,m^{-2}}$) |
| LE_ISBA_avg | Average of the total latent heat flux ($\mathrm{W\,m^{-2}}$) |
| LEI_ISBA_avg | Average of the sublimation latent heat flux ($\mathrm{W\,m^{-2}}$) |
| SWD_ISBA_avg | Average of the downward shortwave radiation ($\mathrm{W\,m^{-2}}$) |
| TS_ISBA_avg | Average of the surface temperature (K) |
| TS_ISBA_max | Maximum daily surface temperature (K) |
| RAM_SONDE_avg | Average of the penetration of ram resistance sensor (m) |
| WET_TH_avg | Average of the thickness of wet snow at the top of the snowpack (m) |
| REFROZ_TH_avg | Average of the thickness of refrozen snow at the top of the snowpack (m) |
| PSN_ISBA_avg | Average of the snow fraction ($-$) |
| TALB_ISBA_avg | Average of the surface total albedo ($-$) |
| DSN_T_ISBA_vcs | Value at the current time step of the total snow depth (m) |
| SNOW_SAT_avg | Average of the snowpack saturation ($-$) |
| COLD_CONTENT_vcs | Value at the current time step of the cold content ($\mathrm{J\,m^{-2}}$) |

than the average over the next 24 h, so it was used in its place. Lastly, information from the top soil layer was also retrieved and aggregated accordingly. The final aggregated Crocus variables are listed in Table 2.

## 2.3 ML-based modelling setups

Besides Crocus, this study compares the performance of three modelling setups; the two hybrid setups that integrate Crocus outputs into a ML framework (PPC and AUG) and a ML model purely based on meteorological measured data. The goal was

to obtain SWE in daily time steps, predicting it forward in time. A general overview of the three ML-based modelling setups is shown in Figure 1.

### 2.3.1 Measurement-based ML model

In MSB, the predictors are the SWE value in the current daily time step and the daily averaged meteorological features for the 24 h before the next one. Lagged meteorological variables for the previous 14 days were included to account for delayed snowpack

responses to atmospheric conditions. The variable predicted by the ML model is the corresponding $\Delta$SWE, calculated as the difference in SWE between next and current daily time steps. It should be noted that our approach limits model training to




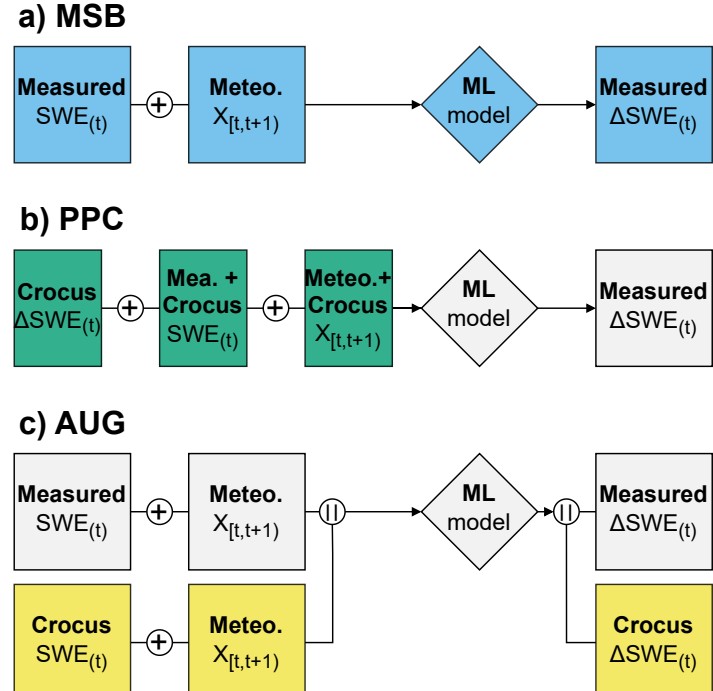

**Figure 1.** Diagram showcasing the training setups for the a) measurement-based, b) post-processing and c) data-augmentation approaches. MSB is represented in blue. For PPC and AUG, the shared elements with MSB are coloured in light gray and the differences are highlighted in green and yellow, respectively. The boxes represent all samples available for training except for the yellow ones, which correspond to locations not included in training where Crocus was run. The addition symbol represents concatenation of features, and the double vertical bar represents concatenation of observations. The target is defined as $\Delta\text{SWE}_{(t)} = \text{SWE}_{(t+1)} - \text{SWE}_{(t)}$. In case of adding lagged variables, the meteorological features are also added for $X_{[t-1,t]}, X_{[t-2,t-1]}$, etc.

stations where consecutive daily SWE measurements are available, that is, the automatic stations. Furthermore, to avoid a bias towards zero in the predicted variable due to long periods without snow cover, measurements for which the next time step has zero SWE were removed. Ultimately, the number of available samples was 1874.

**2.3.2 Post-processing hybrid model**

In PPC, the physically-based model target is given as an input to the ML model to produce a corrected or post-processed version of it. In practice, this setup is implemented similarly to MSB, but the ML model is given additional predictors. The key addition is the daily $\Delta\text{SWE}$ retrieved from the Crocus simulations, the target to be post-processed. The current SWE value according to the Crocus simulations is also included to complement the measured one. Additional Crocus-based predictors,



such as the ones described in Table 2, may also be added on top of the meteorological features. The rationale is that the ML model can rely on the physical information stored in the Crocus simulations as a base for its prediction, correcting it when necessary based on the meteorological information.

### 2.3.3 Data-augmentation hybrid model

The AUG setup also resembles MSB, except that the training dataset has been augmented with synthetic observations. These
additional ΔSWE samples were derived from the Crocus simulations at the manual stations, which were not included in training. This corresponds to an additional 18717 samples. To balance the influence of the Crocus-generated data on ML model training, they are given a smaller weight in the loss function computed as the ratio of the number of measured to augmented training samples. There are two ways of interpreting this approach. First, as a measurement-based ML model that is implicitly regularized by adding model-generated data to guide its training. Second, as an ML surrogate of Crocus to which
we incorporate SWE observations, but where the training for both observed and modelled data is performed simultaneously, simplifying the process.

### 2.4 ML Model selection and evaluation

The process of ML model selection, training and evaluation followed the same general steps for the three ML-based modelling setups. First, the data was split into three sets: train, validation and test. Then, a ML model was initialized for each algorithm
and hyperparameter combination and was fitted to the train set. Three different ML algorithms were compared: a random forest (RF), implemented with the scikit-learn library (version 1.3.0., Pedregosa et al., 2011), a feed forward neural network (NN) and a long-short term memory neural network (LSTM), implemented in the Keras library (version 2.12.0, Chollet et al., 2015). The available hyperparameter combinations are described in Appendix A. After that, the models were used to predict in the validation set, from which the mean squared error of ΔSWE was computed. All except the model with the lowest error
were dismissed. Finally, the winning model was re-trained with both training and validation data, and used for prediction in the test set for a final evaluation. This process was independently applied to two data partitioning strategies, named temporal and station splits, to assess the robustness of each approach for forecasting at locations with and without historical SWE measurements, respectively. By employing these distinct split types, models were optimised for their individual predictive goals in a data-efficient way.

In the temporal split, represented in Figure 2a, a leave-one-out cross-validation strategy was used involving five contiguous folds of approximately 20% of each station's data. Each fold begins and ends roughly at the beginning of the hydro-year, ensuring that they contain at least one year of measured data. For each split in the cross-validation loop, a separate model is created that uses three folds as train set, one for the validation set and one as test set. In AUG, all models are also trained on the full time series of Crocus simulations at the seven manual stations during both model selection and evaluation. The station split,
represented in Figure 2b, again follows a leave-one-out cross-validation strategy, but in this case the full time series from two of the automatic stations conform the train set and the remaining station is used as validation set. The test set is comprised of all data available at the seven manual stations. In AUG, another leave-one-out cross-validation loop is added after model selection





to avoid testing the model on stations contained in the Crocus simulations employed for its training. So, the augmented data consists only of six manual stations, and the remaining one constitutes the test set, producing seven models corresponding to
each cross-validation split.

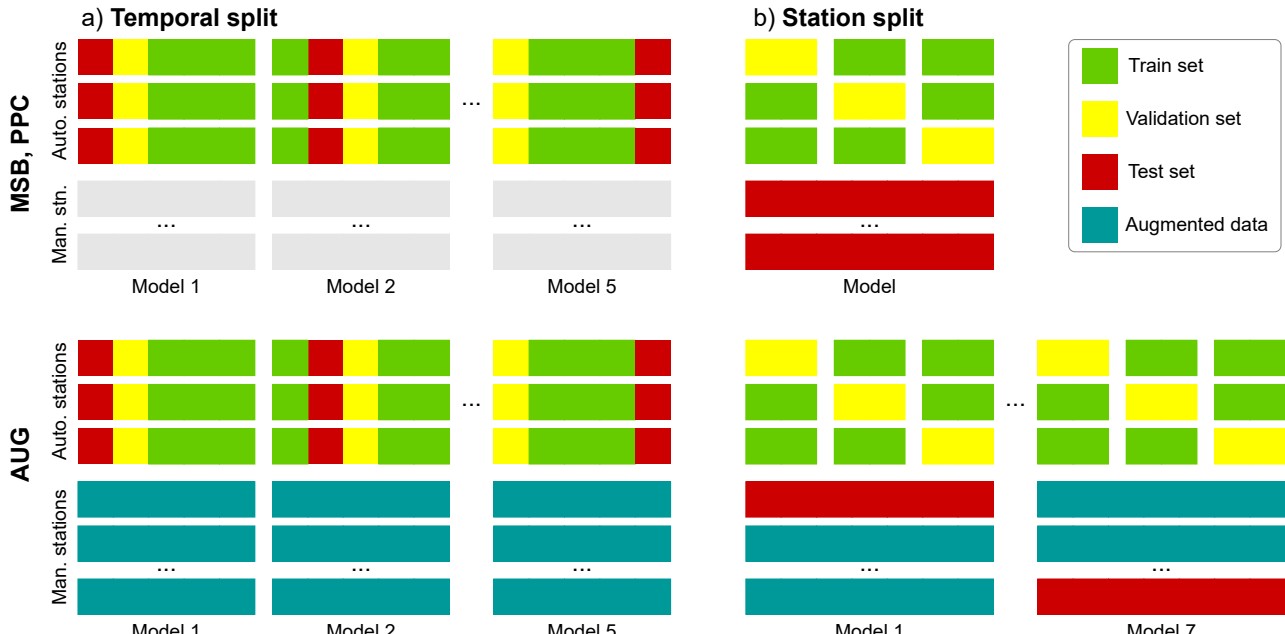

**Figure 2.** Diagram representing the train, validation and test sets used to select, train and evaluate the MSB, PPC and AUG models for a) the station split and b) the temporal split strategies. Each rectangle represents the full time series at a given station, the first three rows represent the automatic stations and the remaining ones the seven manual stations. The augmented data is also represented, as it is part of the training set, but uses the Crocus simulations instead of measured data.

## 2.5    Analysis of SWE predictions

To evaluate the performance of the modelling setups for predicting SWE, the trained models were employed to generate a single time series of SWE for each modelling setup and station. First, an initial condition of SWE = 0 was set at the starting date. Then, ΔSWE was predicted using the model whose test set encompasses that time step and station. Next, the predicted
ΔSWE was added to the previous SWE, replacing any negative SWE values by zero. Lastly, the updated SWE value was stored and used as input for the subsequent date. This process was performed iteratively until input data is no longer available.

The data points for which there is measured SWE data were used to assess the model performance. The metrics computed for this study were the root mean squared error (RMSE), mean bias, and Nash-Sutcliffe efficiency (NSE). Furthermore, the feature importances were retrieved from each of the ML models for their test predictions using the SHAP library (Lundberg
and Lee, 2017), which quantifies the impact of the predictors on the model output for each time step.





# 3 Results

## 3.1 Optimal ML model configuration

The results of model selection revealed the Random Forest algorithm to be consistently superior in all modelling setups. The NN and LSTM models attained 13% and 23% higher MSE values than RF on average, for all setups and splits. The differences

in performance for different RF hyperparameters were not large, with differences below 5% in MSE at the most. No clear positive or negative trend was found for any hyperparameter, besides generally getting slightly better results for larger values of the number of features. The hyperparameter configurations used for each modelling setup and split type are reported in Table 3. The other hyperparameters were left at their default value. For the remainder of this section, only the results from the best performing ML algorithm and hyperparameter combination for each setup are presented. Training was performed in under a

minute for the RF models, using a single CPU core. Inference time for a single time step is on the order of magnitude of 50 ms. The results shown in the following sections used lagged meteorological variables, but not the additional Crocus variables for PPC, which yielded the best performance. The impact of these two modelling choices is further elaborated in Section 3.4.

**Table 3.** Optimal Random Forest hyperparameter configurations for the different ML-based setups and split types.

| Split Type | Hyperparameter | MSB | PPC | AUG |
|---|---|---|---|---|
| Temporal split | Max Depth | 10 | 10 | None |
| | Max Samples | None | 0.5 | None |
| Station split | Max Depth | 10 | 10 | 20 |
| | Max Samples | None | None | None |

## 3.2 Predicted SWE comparison

The test set NSE on the corresponding stations for each data split is displayed in Figure 3 for all modelling setups. The most

noticeable difference is the large gap in performance in all models between prediction on a) left-out years at the automatic stations and b) extrapolation to the manual stations. These correspond to the models trained and evaluated using the temporal and station data splits, respectively. In the temporal split, all models achieved similarly good performances, roughly between 0.85 and 0.95 test NSE. All ML-based setups outperformed Crocus, PPC achieving the highest score. AUG, however, failed to surpass the performance of MSB. Despite the success of the ML-based setups, they consistently underperformed Crocus at the

two stations with smaller sample size. In the station split, all model performances decreased significantly, especially for MSB and PPC. These setups obtained a test NSE slightly below 0.70, but most stations remained below 0.3 and some even reached negative NSE values, indicating that the variation of the error was equal or larger than the variation in the observed data. In contrast, AUG attained the best performance with a test NSE of 0.85, even higher than the 0.80 of Crocus. Moreover, AUG exhibited the least performance variability across stations, none of them reaching below 0.46 NSE.





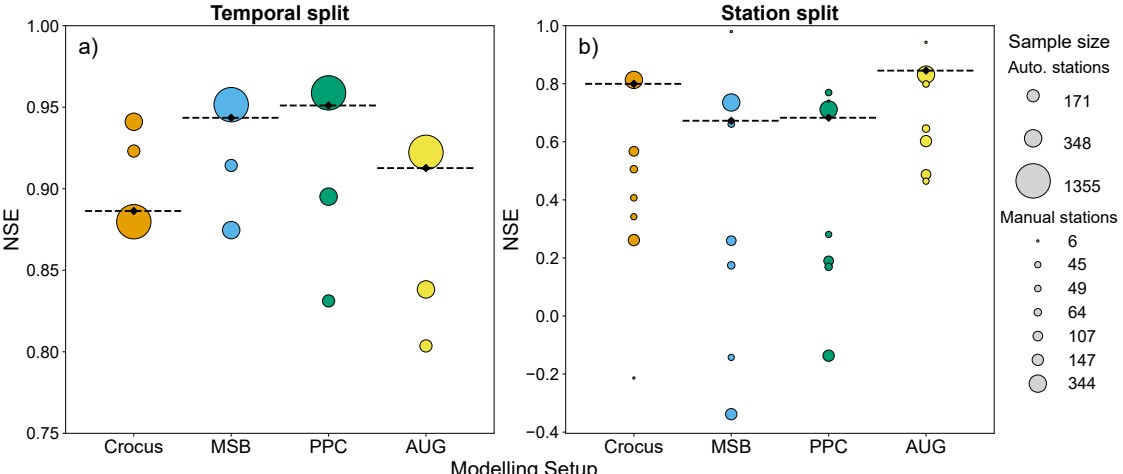

**Figure 3.** Bubble plots showing the NSE achieved by using each modelling setup to predict SWE in forward simulation for the data-rich stations in the temporal split (a) and the test stations in the station split (b). It is important to note the change in y-axis scale between the two panels. The size is proportional to the number of samples used for computing the NSE on each station, and the horizontal lines represent the NSE evaluated on all stations.

### 3.2.1 Time series of automatic stations

Figure 4 shows an example five-year subset of the SWE time series at one of the automatic stations. All four modelling approaches display good agreement with the observed snowpack dynamics; that is, the seasonal snow pattern is generally well reproduced. Moreover, they succeed in capturing the variability in yearly peak SWE values reasonably well, with deviations of 10-15% on average compared to the measured values. However, AUG and Crocus tend to underestimate the amount of snow. This argument is also supported by the mean bias at the test stations (Appendix B), which is more than two times smaller for MSB and PPC than for the other setups. Because of this, the peak SWE values are also underestimated, especially by Crocus. Another noticeable pattern is the increase in absolute error of the predicted SWE as the snow cover period advances. Importantly, a significant spike in the residuals is observed at the last part of the snow ablation period due to the inability of the models to accurately reproduce its exact timing. Once more, this effect is most pronounced in Crocus, which predicts the full melt almost 6 days earlier than measured, on average. Conversely, the ML-based counterparts exhibit only a slightly positive shift in the snowpack melt-out date, most pronounced in the PPC mode, which results in smaller error overall.

### 3.2.2 Time series of manual stations

To compare the predicted SWE time series of the models trained with the station split, Figure 5 shows a five-year fragment from the manual station with most available samples. The first noticeable characteristic is a much poorer fit, relative to the predictions in automatic stations. In particular, the residuals grow significantly after the main snow accumulation phase due to a large underestimation of SWE. This bias has a considerable effect on the peak SWE values, which have a more than 100





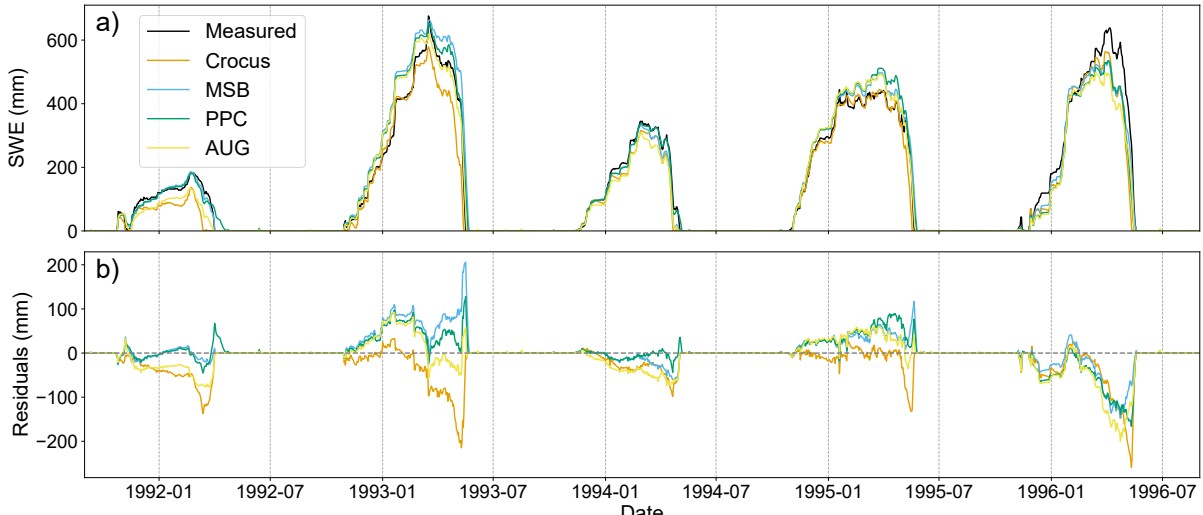

**Figure 4.** Time series for an example five-year range in the Reynold Mountain East station, the automatic station with most available samples, showing the SWE time series (a) and the corresponding residuals (b) from the measured data, Crocus SWE simulations, and predicted by the ML-based setups trained with the temporal split.

mm deficit on average for MSB and PPC and around 80 mm for AUG. While Crocus still has large absolute errors in peak SWE prediction, it can predict it with much fewer bias, around 25 mm. This trend is also reflected in a large increase of the test mean bias for the ML-based setups compared to the automatic station predictions (Appendix B), which now obtain much larger

values than Crocus, especially MSB and PPC. However, AUG has less absolute mean bias for the majority of the stations, and improves by more than 10% its test RMSE. Notably, AUG can predict the snow ablation much better than any other setup, achieving lower residuals in the last few snow measurements for each snow year in almost all instances displayed in Figure 5, and more generally across all stations and years with sufficient available measurements. It is important to note that the models have much larger variability in performance between stations and years, making it difficult to find a subset that represents the

whole test set well.

### 3.3 Feature importance of the ML-based models

According to the results of the SHAP analysis for both types of split (Figure 6), the most important variable for most setups was the downwards shortwave radiation. The positive integral of the air temperature was also amongst the three most important features, indicating that the energy balance highly influences the predictions of the models. The observed SWE obtained the

second largest mean value across the ML models, which is reasonable since this is the only variable which contains direct information on the state of the snowpack. Interestingly, PPC gives very little importance to this variable, despite obtaining the best performance in the temporal split. Finally, the snowfall also features in the top five variables as the strongest positively correlated variable with ΔSWE. It is important to note that the post-processing approach also gives significant importance to





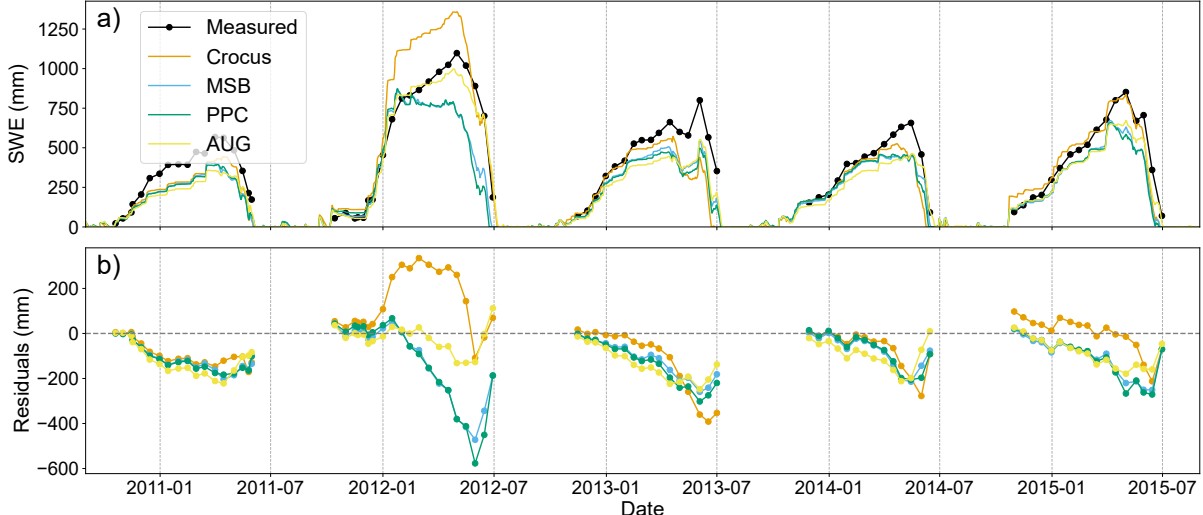

**Figure 5.** Time series of a five-year range in the Weissfluhjoch station, the manual station with most available samples, showing the SWE time series (a) and the corresponding residuals (b) from the measured data, Crocus SWE simulations, and predicted by the ML-based setups trained with the station split. Because of the large portion of missing data, the measured SWE and derived errors are displayed with dots connected by line segments for better visibility.

the change in SWE simulated by Crocus, reaching above a SHAP value unit. This shows that the modelled target is indeed a

valuable variable for SWE forecasting, although far from the most important. Other variables that obtained a mean absolute SHAP value higher than 0.5 include both aggregate variables regarding to air humidity and downwards longwave radiation. The SHAP analysis for the temporal (a) and station (b) splits share many similarities. Indeed, the top five most important variables are exactly the same. However, there are some interesting changes. The most noticeable is a significant increase in the importance of the observed SWE in PPC for the station split. Additionally, less importance is given to the short-wave

radiation, especially in MSB, and slightly more to the air temperature.

     The ML models were able to capture some expected physical relationships, such as negative correlation with the air temperature and shortwave radiation and a positive one with snowfall. Moreover, when predicting on the training stations with the temporal split (Figure 7a-c) the snowfall rate was found to have the expected linear relationship with the predicted output. There was only some bias for higher snowfall values, which was corrected in AUG. Nevertheless, for the ML-based models

tested using the station split (Figure 7d-f), this bias is even more evident. So, while there was still a clear linear relationship between SHAP and observed snowfall for low to mid values, above a threshold of approximately $0.7 \, \mathrm{g \, m^{-2} \, s^{-1}}$, any additional snowfall did not result in a further increment of SWE. This indicates that the ML models could not extrapolate to stations with higher snowfall events, and underpredicted these extreme cases.



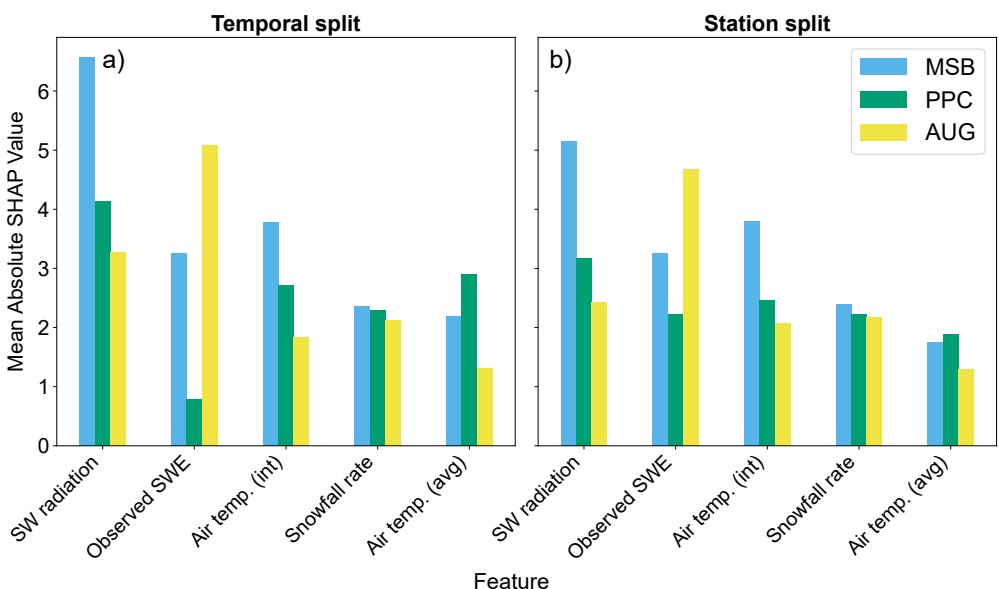

**Figure 6.** Feature importances of the five most influential variables on the output of the ML-based models, aggregated for all lagged variables for the temporal (a) and station (b) splits. They are ordered according to their average importance for the three ML-based setups.

## 3.4 Impact of modelling choices

### 3.4.1 Lagged feature engineering

The impact of adding lagged meteorological information in the ML-based model inputs was tested by comparing the results in which the ML models were given the previous 14 days of meteorological information as additional inputs against the same models without any lagged variables. The reported RMSE of both and the improvement from one to the other is shown in Table 4. The RMSE obtained with Crocus is shown as well for reference. In the temporal split, the lagged version reduced the error of MSB and AUG significantly, while PPC achieves similar results. For the station split, a similar pattern is observed. This time the decrease in RMSE for MSB and AUG is even larger, roughly 37%, while for PPC it remains around 5%. Hence, having Crocus-simulated variables as input makes the model much less dependent on past information to obtain good performance, which becomes especially relevant when predicting on new stations. The reason may be that the Crocus predictors already implicitly include the memory of the past days.

Moreover, the importance of the added lagged features was explored through an analysis of their SHAP values. Very low SWE values had a positive effect on the predicted ΔSWE, but otherwise its net effect was negative. The impact of this variable fluctuated greatly, though, showing the strongest interaction effect with the air temperature. Figure 8 shows the mean absolute SHAP value of the 14 lagged values for the three most influential meteorological variables, determined in Section 3.3. In general, feature importance decayed very quickly for larger lag values, but there was some distinction between features. The





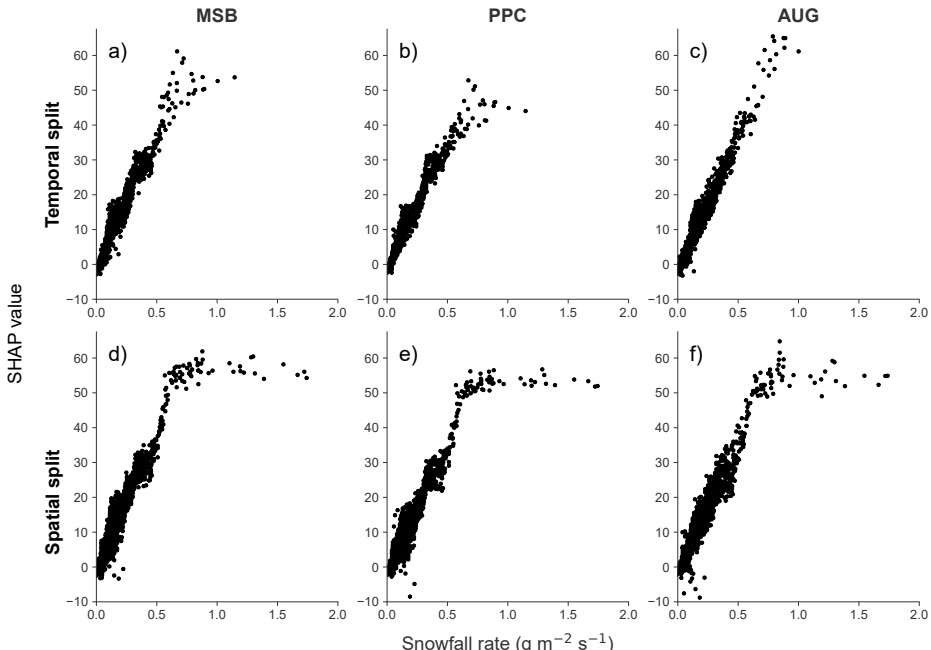

**Figure 7.** Scatter plot of the SHAP values against measured values of the daily averaged snowfall rate. The SHAP values quantify the deviation in the predicted ΔSWE from its average value over the test samples, as caused by the specific value of the snowfall rate in each of them. The three ML-based setups are compared for both temporal (a-c) and spatial (d-f) data splits.

**Table 4.** Comparison of RMSE values for each setup and split with and without adding lagged variables, and the percentage difference (Diff) between them.

| Split | Lag | Crocus | MSB | PPC | AUG |
|---|---|---|---|---|---|
| | No | 55.1 | 48.3 | 38.5 | 56.4 |
| Temporal | Yes | - | 38.9 | 36.2 | 48.3 |
| | Diff | - | -19.5% | -6.0% | -14.3% |
| | No | 124.3 | 255.4 | 164.8 | 173.6 |
| Spatial | Yes | - | 159.0 | 156.5 | 109.3 |
| | Diff | - | -37.8% | -5.1% | -37.1% |

shortwave radiation had the softest decrease in importance, with some of the lagged inputs within the preceding week having a noticeable effect. For AUG in particular, the relevant lagged window was further reduced to 3 days. The air temperature, only the value at the day before had a relevant impact, which was mostly null for larger lagged values. Lastly, the snowfall rate of





any of the previous days had little to no impact on the model predictions. In conclusion, the addition of lagged meteorological variables was not relevant for more than a week before, and in some cases markedly less.

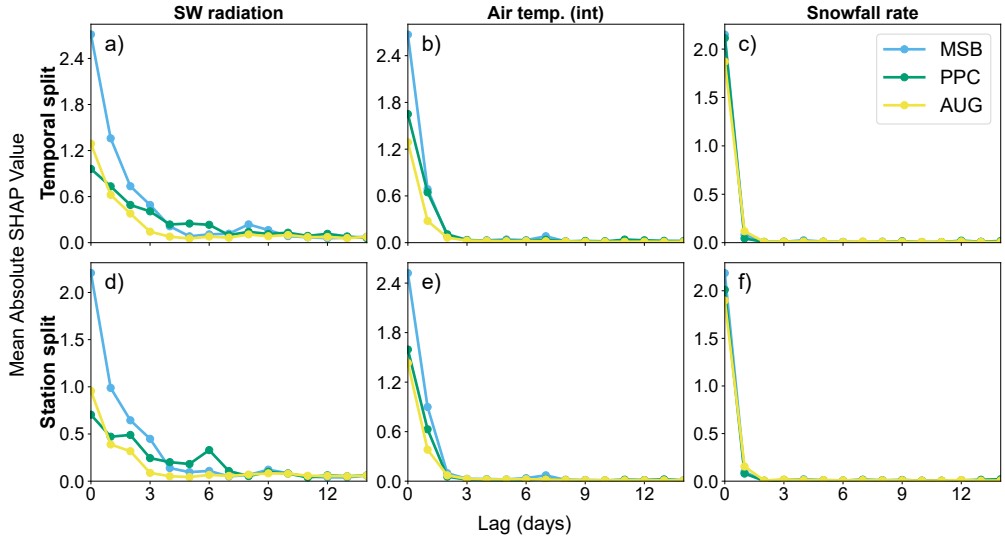

**Figure 8.** Feature importance of the 14 historical values of the three most influential meteorological variables for the temporal (a-c) and station (d-f) splits.

### 3.4.2 Crocus feature engineering

To determine the usefulness of incorporating further Crocus state variables for SWE prediction, another PPC model was trained that also included the inputs defined in Table 2. By providing additional context on the state of the modelled snowpack, the ML model is expected to have information available for better correcting the Crocus predictions. When trained with the temporal split, the enriched PPC model achieved an NSE of 0.95, the same as the regular PPC. Hence, despite the good performance, the simpler model was favoured for the analysis. For the station split, the NSE dropped to 0.20, substantially lower than any of the other setups. In four out of seven test stations it yielded negative NSE values up to -3.00. When investigating the variable importances of the enriched PPC model, the main difference amongst the most influential variables was the replacement of the downwards shortwave radiation by the Crocus-reported average net radiation, which in turn gained even more importance in the model predictions.

### 4 Discussion

The results of the study show that hybrid models can outperform both state-of-the-art numerical snow models and classic ML approaches for SWE simulation at point locations, based on meteorological data. However, the optimal type of hybrid setup highly depends on the intended use of the model.





When predicting at locations for which historical SWE measurements are available for model training, all ML-based models outperformed Crocus. These results are in line with recent literature in ML for hydrology, which has been found to outperform traditional numerical models in a variety of tasks (Mosaffa et al., 2022). The differences in performance concentrate towards the end of the snow period, where the ML-based models particularly improve the timing of the snow melt. In particular, the hybrid PPC setup achieves the best performance in all metrics computed in this study. The improvement of this type of hybrid setup over traditional ML approaches for SWE prediction coincides with the findings of Steele et al. (2024), who demonstrated that a similar PPC strategy outperformed a MSB counterpart as well as other statistical and physically-based models for predicting SWE and snow density using similar predictors. On the other hand, AUG was found to perform worse than MSB, indicating limited transferability of the snow dynamics between the stations simulated with Crocus and those with measured SWE data. Finally, all ML models did capture expected correlations between ΔSWE and its predictors. For instance, downward short-wave radiation and air temperature were amongst the most influential variables dominating snow melt, and there was a clear positive linear correlation between snowfall and increase in SWE. This suggests that such models are able to correctly identify physical patterns in the data when the training data is sufficiently representative of the application domain.

When predicting on new locations not present in model training, the performances of all setups were lower and had larger variability than in the temporal split. MSB achieved the worst performance, mostly due to a strong negative bias. These results are consistent with a well known limitation of machine learning models, namely their need for large, representative datasets (Xu and Liang, 2021). PPC obtained similarly poor results, indicating that the Crocus corrections learnt by this type of hybrid setups also do not generalize well to other stations. However, AUG succeeded in improving Crocus, reducing its RMSE by more than 10% and reaching higher NSE at all but one station, although it did exhibit a greater underprediction bias. This success could be attributed to Crocus having a greater generalisation capability to different regions, so the ML model in AUG is able to transfer its knowledge when predicting in stations with similar meteorological conditions. Nonetheless, all ML-based models displayed impoverished physics behaviour compared to the temporal split; for instance, the predicted increase of SWE preserved the linear behaviour with snowfall for mid to low values but saturated after a certain snowfall value was surpassed. However, AUG could extrapolate slightly further than the other setups, which could indicate that targeting more extreme meteorological conditions in the augmented data could reduce this type of anomalies in this setup.

An analysis on the importance of lagged variables shows that their addition significantly improves the results for most methods except for PPC, which displayed only minor improvements. This could be expected given that Crocus already digests the information from the previous meteorological conditions affecting the snowpack, hence its output already contains lagged information implicitly that the ML model can exploit. Nevertheless, the importance of the lagged variables rapidly decays with the number of lagged days and is essentially null after a week, which suggests that longer lag times may not be needed. Incorporating lag was most impactful for the downwards shortwave radiation, likely in link with snowpack warming and ripening prior to melt. An additional experiment was performed to investigate the effect of introducing Crocus state variables as additional predictors to the PPC setup, which resulted in little improvement in the temporal split and a large performance loss for the station split. The latter seems to be caused by an over-fitting of Crocus variables such as its reported net radiation,





which might not be as generalizable to other stations as similar measured variables. These results should serve as a warning against indiscriminate variable inclusion, emphasizing the need for a more judicious selection of input variables.

## 4.1 Limitations and recommendations

The main limitation of this study is the small number of measured locations. The choice of this data set was motivated by the high quality of the data, consisting of *in-situ* SWE and meteorological measurements for 7 to 20 years, and the diversity in geographical locations. However, only ten stations were available, of which only three had daily SWE measurements. This is especially severe for the station split; the hyperparameter tuning was performed by training on only two stations and validating on a third, and for evaluation it was trained on the three stations and evaluated on the remaining seven. Moreover, the forecasted SWE in the test stations could only be compared to sparse SWE measurements, but not with the target variable, daily $\Delta$SWE. Hence, additional research on larger datasets would be recommended to provide a complete assessment of the methods described in this paper, which may need to rely on derived SWE products, given the small size and spatial scope of measured SWE datasets. Another important point is the known errors of the SWE and meteorological datasets, acknowledged in Ménard et al. (2019). Given the data limitations, deviations in the data could become specially relevant. Furthermore, this study did not quantify the uncertainties of the SWE predictions.

These hybrid setups are especially interesting when considering large or global scale SWE predictions. A crucial improvement of the AUG setup over Crocus and other similar physically-based models in this regard is that it has shorter inference times. Moreover, Crocus requires a detailed set of meteorological variables that may not be available at larger scales or for future scenarios, whereas the ML-based nature of this approach enables adjusting to any available predictors. On the other hand, PPC is expected to improve its performance when trained on larger datasets, since it would broaden the interpolation range of the model, but it still relies on running the physically-based model on all training and inference station-years. Therefore, testing simpler and faster physically-based models for this setup in future studies would be highly relevant. Larger training sizes could also affect the optimal choice of ML algorithm, instead of the current RF. For example, other studies have shown that LSTMs can result in enhanced SWE simulations when using larger datasets (Steele et al., 2024; Duan et al., 2024; Cui et al., 2023). In the present investigation, the goal was predicting the change in SWE given only the current state and recent meteorological information, without explicitly accounting for its previous history, but different implementations of LSTM or alternative ML models could result in additional improvements.

## 5 Conclusions

This study tested two hybrid ML approaches for forecasting daily SWE both in left-out years from the training stations and in independent test stations. Data from ESM-SnowMIP including *in-situ* measurements of SWE and meteorological data from ten stations throughout the northern hemisphere was used for this purpose. The first approach followed a commonly used hybrid implementation in which the output simulated by a physical snow model, Crocus, was taken as an additional predictor to the meteorological data. This setup outperformed both Crocus and a ML model based only on meteorological data for predicting





in left-out years, suggesting that ML models can benefit from additional model-simulated information. However, when tested on the independent stations, this setup performed significantly worse than Crocus, indicating that the knowledge gained in the training stations could not be generalized to other locations. The second approach involved a novel hybrid setup in the context of SWE prediction, where a ML model was trained not only on measured data but also on Crocus SWE simulations at other stations. This setup failed to improve the ML results for predicting in trained stations, but excelled at prediction in additional

locations, not only significantly improving the results from other ML-based setups, but also reducing the RMSE from Crocus by more than 10%. These results demonstrate that hybrid models, in particular the data augmentation setup, have the potential to produce detailed SWE forecasts at large geographical scales by using physically-based model simulations to complement the information provided by observed data.

**Appendix A:  ML hyperparameter choices**

Three model types are explored in this study: random forest (RF), fully connected neural networks (NN) and long-short term memory (LSTM). The different hyperparameters tested for each of them can be found in Table A1. For RF, two parameters are tuned: the maximum depth of the trees that conform the RF algorithm (`max_depth`), which allows to find the right balance between bias and variance; and the subsample size for each tree (`max_samples`), which allows to find the right balance between stability and diversity of the trees. For both NN and LSTM models, three parameters were tuned: the model's

architectured, which determines the number of layers and their units (`layers`); the learning rate (`learning_rate`), which determines the step size during weight optimization, crucial for converging to a global minimum; and the strength of the L2 regularization (`l2_reg`), which can prevent overfitting and improve generalization of the models. A range from simpler to more complex architectures is tested with the goal of improving model performance.

**Table A1.** Model types and hyperparameter choices that are compared for each setup.

| Model | Hyperparameter | Choices to test |
|-------|----------------|-----------------|
| RF | max_depth | None, 10, 20 |
|    | max_samples | None, 0.5, 0.8 |
| NN | layers | [2048], [128, 128, 128] |
|    | learning_rate | 1e-2, 1e-4 |
|    | l2_reg | 0, 1e-2, 1e-4 |
| LSTM | layers | [512], [128, 64] |
|    | learning_rate | 1e-2, 1e-4 |
|    | l2_reg | 0, 1e-2, 1e-4 |





## Appendix B: Results of additional metrics

The performance of the models according to their NSE, RMSE and mean bias for both temporal and station splits is reported in Table B1.

*Code and data availability.* The software used for obtaining the results of this study is available at https://github.com/oriol-pomarol/snow_ project. The SWE and meteorological raw data can be accessed from Ménard et al. (2019), and the Crocus snowpack simulations from Lafaysse (2025).

*Author contributions.* Oriol Pomarol Moya, Madlene Nussbaum, Siamak Mehrkanoon, Philip Kraaijenbrink, Derek Karssenberg and Walter W. Immerzeel were involved in the conceptualization and design of the methodology, carried out and analysed by Oriol Pomarol Moya. Isabelle Gouttevin provided part of the data and assisted in the analysis. Oriol Pomarol Moya prepared the manuscript with contributions from all co-authors.

*Competing interests.* The authors declare that they have no conflict of interest.

*Acknowledgements.* We acknowledge the use of the Eejit HPC, supported by the Earth Sciences and Physical Geography departments at Utrecht University, and thank the engineers in charge of its maintenance. During the preparation of this manuscript, the authors used AI tools in order to suggest text reformulations and enhancements, with the aim of improving the language quality. After using this tool/service, the authors reviewed and edited the content as needed and takes full responsibility for the content of the publication.



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





**Table B1.** Model performance metrics across different stations and the full test set for temporal and station splits.

| Split type | Station | Samples | NSE | | | | RMSE | | | | Mean bias | | | |
|---|---|---|---|---|---|---|---|---|---|---|---|---|---|---|
| | | | Crocus | MSB | PPC | AUG | Crocus | MSB | PPC | AUG | Crocus | MSB | PPC | AUG |
| Temporal split | cdp | 348 | 0.94 | 0.87 | 0.90 | 0.84 | 29.0 | 42.3 | 38.7 | 48.1 | -15.6 | -6.0 | 6.0 | -24.9 |
| | rme | 1355 | 0.88 | 0.95 | 0.96 | 0.92 | 62.9 | 39.9 | 36.8 | 50.5 | -31.1 | -10.0 | -8.6 | -21.9 |
| | sod | 171 | 0.92 | 0.91 | 0.83 | 0.80 | 15.8 | 16.7 | 23.5 | 25.3 | 1.2 | -8.5 | -22.1 | 1.4 |
| | TEST | 1874 | 0.89 | 0.94 | 0.95 | 0.91 | 55.1 | 38.9 | 36.2 | 48.3 | -25.2 | -9.1 | -7.1 | -20.3 |
| Station split | oas | 49 | 0.41 | 0.66 | 0.77 | 0.80 | 20.8 | 15.7 | 13.0 | 12.1 | -10.2 | 2.8 | -1.2 | -7.7 |
| | obs | 45 | 0.34 | -0.14 | 0.28 | 0.46 | 19.3 | 25.4 | 20.1 | 17.4 | 0.0 | 7.8 | 7.1 | 6.5 |
| | ojp | 64 | 0.51 | 0.17 | 0.17 | 0.65 | 17.9 | 23.1 | 23.2 | 15.1 | 8.7 | 12.3 | 12.6 | 8.2 |
| | sap | 6 | -0.21 | 0.98 | 0.74 | 0.94 | 77.9 | 10.0 | 36.1 | 16.9 | -43.2 | 2.6 | -12.7 | -3.7 |
| | snb | 107 | 0.57 | 0.26 | 0.19 | 0.49 | 126.4 | 165.3 | 172.9 | 137.5 | 46.2 | -47.3 | -57.4 | -33.4 |
| | swa | 147 | 0.26 | -0.34 | -0.14 | 0.60 | 191.5 | 257.7 | 237.5 | 140.5 | -164.6 | -220.9 | -206.2 | -114.2 |
| | wfj | 344 | 0.81 | 0.74 | 0.71 | 0.83 | 115.5 | 137.4 | 143.6 | 109.7 | -19.6 | -93.9 | -98.7 | -75.0 |
| | TEST | 762 | 0.80 | 0.67 | 0.68 | 0.85 | 124.3 | 159.0 | 156.5 | 109.3 | -34.4 | -89.9 | -91.1 | -60.0 |

NSE: Nash-Sutcliffe efficiency, RMSE: root mean square error in mm w.e.