# Peer review of "Improving forecasts of snow water equivalent with hybrid machine learning"

_EGUsphere, 2025_

## Referee Comment (RC1)

Review of "**Improving forecasts of snow water equivalent with hybrid machine learning**"

Pomarol Moya et al.

This manuscript presents a hybrid ML approach that combines SWE and meteorological observations with output from a physical snow model to provide enhanced SWE forecasting. I believe these types of hybrid approaches represent an exciting future for snow modeling. The manuscript is well written and the figures are generally clear. However, I have some major concerns with this approach, especially regarding the inclusion of in-situ meteorological data and how this impacts the application of this approach to global SWE forecasting, which is described by the authors as a main motivation for this work.

**Major concerns**

My main concern is that I struggle with the application of this approach. In the abstract, the authors say "potential to improve forecasts of SWE at unprecedented spatio-temporal scales". However, in the manuscript the ML models are tested only in areas with weather station data and the in-situ meteorological data at times $t$ and $t+1$ (forecast time) are used as input features for the model. This severely limits the applicability of the model from a forecasting perspective (in-situ meteorological conditions are not available at time $t+1$) and to specific locations with in-situ meteorological data. Ideally for a large-scale SWE forecast application, such a model would be applied with meteorological output from a model forecast. However, this manuscript does not evaluate how such an approach would perform for this application. In its current form, the methods and models presented in this manuscript are limited, and I don't believe they have much use for forecasting SWE, especially at the global scale as only a handful of sites are utilized in this study.

**Methodological comments**

L72 - Why were only 10 stations utilized? It seems that this approach could benefit greatly from an increase in training data and there are certainly more stations in the NH with timeseries of SWE data that could be used for training. Even if a site has data from only a few years surely this would still be useful, no?

L85 – "... according to the geographical location of each station." What does this mean? Where different aggregation methods used in different locations?

Table 1 – Why do you use both SWdown_avg and SWdown_day? These features will be nearly perfectly correlated and I doubt both are necessary.

Table 2 – What is RAM_SONDE_avg and why is it a useful feature for the ML?

Figure 1 – The formatting for this diagram is a bit confusing. Why are [ and ) brackets used? I also think that $\Delta SWE_{(t)}$ is confusing. I see that it is defined in the caption, but it is not immediately intuitive as really the target variable is the change in SWE at time *t+1*. Also Measured is shortened to Mea. In b) but not a) or c).

L125 – "Additional Crocus-based predictors, such as the ones described in Table 2, may also be added…" What is meant by this? More details are necessary on this.

L140 – "Three different ML algorithms were compared:" Why were these three chosen? How was the LSTM model set-up? It's not surprising that the LSTM does not perform optimally as these are typically better with longer time series of data. Perhaps a GRU model would be preferable? For the NN and the LSRM, how were the hyperparameters tuned? RFs typically can perform better 'out of the box'. In contrast NNs typically require much more substantial hyperparameter tuning. From Table A1 it's not surprising that the NN and LSTM did not perform as well as it seems that not very many hyperparameters were tested.

L168 – "Nash-Sutcliffe efficiency" I'm not immediately familiar with this metric. Maybe explain briefly?

**Feature importance**

To me, it is not expected that downwards shortwave radiation would be the most important feature. You are modeling both the accumulation and ablation season correct? I would expect SW radiation to be very important but only during the ablation season. I'm curious if the feature importances change temporally? This might be interesting insight to include. My guess is that SW radiation has high magnitude SHAP values during the ablation season because $\Delta SWE$ is generally much higher during the ablation than the accumulation season. I'm curious what you would see if you compute relative SHAP values (by normalizing by $\Delta SWE$). I would expect other features (precipitation) to be relatively more important.

L229 – "reaching above a SHAP value unit." What is meant by this?

Figure 6 – Why did you choose to plot the mean absolute SHAP values? For some features, it may also be interesting to see *how* the feature impacts the predictions (i.e., increasing or decreasing predicted SWE).

**Discussion**

L281/2 – "The differences in performance concentrate towards the end of the snow period, where the ML-based models particularly improve the timing of the snow melt." Does this indicate that there are substantial errors in the melt dynamics in the physical model?

**Technical comments**

Mind consistency with 'an ML model' vs. 'a ML model' (for example in the abstract both are used). I personally don't know which is correct but try and be consistent with your usage!

Table 2 – Type "soild" in row 3

L249/250 – Different tenses are used in the same sentence here ('reduced', 'achieves').

L260 – Maybe 'slowest' instead of 'softest' here?

L300 – I'm not sure that 'impoverished' is the best word choice here. Maybe just 'poor' is better.

---

## Author Comment (AC1)

Review of "**Improving forecasts of snow water equivalent with hybrid machine learning**"

Pomarol Moya et al.

This manuscript presents a hybrid ML approach that combines SWE and meteorological observations with output from a physical snow model to provide enhanced SWE forecasting. I believe these types of hybrid approaches represent an exciting future for snow modeling. The manuscript is well written and the figures are generally clear. However, I have some major concerns with this approach, especially regarding the inclusion of in-situ meteorological data and how this impacts the application of this approach to global SWE forecasting, which is described by the authors as a main motivation for this work.

Dear reviewer, thank you for your kind comments and also for raising some interesting points of discussion about our manuscript. We hope to provide a comprehensive answer to them in the following text.

**Major concerns**

My main concern is that I struggle with the application of this approach. In the abstract, the authors say "potential to improve forecasts of SWE at unprecedented spatio-temporal scales". However, in the manuscript the ML models are tested only in areas with weather station data and the in-situ meteorological data at times t and t+1 (forecast time) are used as input features for the model. This severely limits the applicability of the model from a forecasting perspective (in-situ meteorological conditions are not available at time t+1) and to specific locations with in-situ meteorological data. Ideally for a large-scale SWE forecast application, such a model would be applied with meteorological output from a model forecast. However, this manuscript does not evaluate how such an approach would perform for this application. In its current form, the methods and models presented in this manuscript are limited, and I don't believe they have much use for forecasting SWE, especially at the global scale as only a handful of sites are utilized in this study.

Our conclusions may have been articulated more ambitiously than the scope of the paper permitted, so we propose to rephrase the relevant text in the introduction and conclusions to avoid overpromising, and to raise this point more explicitly in the discussion.

Nevertheless, we believe that the paper still adds significant value to the community and is therefore worth publishing. Admittedly, modelled forecasts of meteorological data would be required for forecasting applications, and that was not tested in our work. However, we believe that evaluating these hybrid setups extensively with higher quality in-situ data is a valuable first step in achieving that goal. Especially, our work highlighted the value of physical model data as a complement to in-situ observations for training a machine learning algorithm. In particular, using it for data augmentation significantly improved its spatial transferability.

**Methodological comments**

L72 - Why were only 10 stations utilized? It seems that this approach could benefit greatly from an increase in training data and there are certainly more stations in the NH with timeseries of SWE data that could be used for training. Even if a site has data from only a few years surely this would still be useful, no?

We agree that 10 stations is a small dataset, as elaborated upon in the Discussion section. A more elaborate justification will be provided in the manuscript, in line with what we outline below.

There were multiple reasons for choosing this dataset, most importantly, its quality. It consists of in-situ SWE measurements at high temporal resolution covering a large diversity in geographical locations and station characteristics. It also contains several in-situ meteorological variables which had been used to generate snow forecasts using Crocus. To the best knowledge of the authors, there are no standardized datasets which satisfy these characteristics, and creating our own or significantly expanding it would be an arduous and time-consuming task.

Furthermore, one of the purposes of this paper is to show the performance of hybrid models under data scarcity conditions, since (even globally) only limited daily SWE measuring stations are available. Lastly, this dataset has been previously used for model intercomparison purposes and is well-known in the field, establishing a controlled setting for evaluating our hybrid setups.

L85 – "… according to the geographical location of each station." What does this mean?

Where different aggregation methods used in different locations?

This fragment refers to the calculation of the daytime average, for which the daytime hours are calculated for every day of the year according to the geographical location of the station. We will rephrase the sentence for improved clarity.

Table 1 – Why do you use both SWdown_avg and SWdown_day? These features will be nearly perfectly correlated and I doubt both are necessary.

The explanation and justification of the predictors will be further outlined in the manuscript. Regarding the variables derived from the shortwave radiation, while both are certainly correlated they only obtain an $R^2$ value of 0.67. This is because the first one reports the average shortwave radiation over the 24h, while the second the average over the hours that fall between dusk and dawn, which depends on the day of the year and latitude. The 24h average is more sensitive to seasonality, while the daytime average more directly encapsules the atmospheric conditions, so both were deemed potentially useful. Lastly, the daytime average is not very important according to the SHAP analysis, so it is unlikely that it has a strong negative effect on the performances of the machine learning models.

Table 2 – What is RAM_SONDE_avg and why is it a useful feature for the ML?

The explanation and justification of the predictors will be further outlined in the manuscript. Regarding the ram sonde variable; as described in the table, this Crocus state variable accounts for the "average of the penetration of ram resistance sensor", which is a cone-tipped metal rod designed to be driven downward into deposited snow or firn (American Meteorological Society – glossary of Meteorology, https://glossary.ametsoc.org/wiki/Ram_penetrometer, last access 11 July 2025). The penetration distance of the rod into the snow of firn for a given amount of force is an indication of one important physical (mechanical) property of the snowpack, namely its hardness, much related to the snow density and microstructure. Both properties have important implications for heat transfer within the snowpack (snow thermal conductivity is typically much related to density, e.g. Calonne et al 2011) and to a certain extent, for snow melt. Therefore, this variable is a good candidate to consider in relation to SWE prediction and snowmelt behaviour.

Calonne, N., Flin, F., Morin, S., Lesaffre, B., du Roscoat, S. R., & Geindreau, C. (2011). Numerical and experimental investigations of the effective thermal conductivity of snow. Geophysical Research Letters, 38(23).

Figure 1 – The formatting for this diagram is a bit confusing. Why are [ and ) brackets used? I also think that $\Delta SWE_{(t)}$ is confusing. I see that it is defined in the caption, but it is not immediately intuitive as really the target variable is the change in SWE at time t+1. Also Measured is shortened to Mea. In b) but not a) or c).

Admittedly, that figure lacks some explanation regarding the use of brackets and parenthesis, which will be added to the manuscript. These refer to the aggregation method; each daily value is computed from the hour corresponding to the prior SWE measurement (t) up to, but not including, the same hour next day (t+1). We will also incorporate the other proposed improvements for the final version.

L125 – "Additional Crocus-based predictors, such as the ones described in Table 2, may also be added…" What is meant by this? More details are necessary on this.

This sentence is indeed unclear and would benefit from re-writing. The meaning is that besides including only the model-simulated SWE as an additional predictor, one could also add other Crocus-generated state variables, such as those described in table 2. This directly relates to the contents of section 3.4.2, where we compare the results with and without these additional variables.

L140 – "Three different ML algorithms were compared:" Why were these three chosen? How was the LSTM model set-up? It's not surprising that the LSTM does not perform optimally as these are typically better with longer time series of data. Perhaps a GRU model would be preferrable? For the NN and the LSRM, how were the hyperparameters tuned? RFs typically can perform better 'out of the box'. In contrast NNs typically require much more substantial hyperparameter tuning. From Table A1 it's not surprising that the NN and LSTM did not perform as well as it seems that not very many hyperparameters were tested.

While we fully agree that testing other ML algorithms such as GRU would be a great addition, the aim of the paper was not to provide a thorough comparison of different ML algorithms as the focus is on comparing different hybrid modelling setups. The three proposed algorithms are amongst the most popular; RF and LSTM have been used for hybrid SWE prediction in the literature (e.g., King et al., 2020; Steele et al., 2024) while a feedforward NN offered an intermediate step in terms of complexity. We considered that a sufficient subset of the available options.

The implementation of the LSTM model will be further expanded in the manuscript. The implementation was done by taking the lag time window (14 days) of meteorological variables as the sequence length where the LSTM units unfold. After, a dense layer takes the outputs of the LSTM layer and any additional variables at the last time step to produce the predicted $\Delta SWE$ from the current step until the next one. When applied for inference sequentially, the same procedure was followed after shifting the time window one day forward and updating the current SWE (which is also a predictor) to the last predicted value.

Finally, we agree that more tuning would likely improve the performance of NN and LSTM, but would also require much higher run times. For this paper we decided to use a fixed budget for tuning, finding a model that strikes a balance between accuracy and usability. The goal was not to claim what algorithm works best, but rather to find a good performing one to test the application of hybrid models. We believe this needs to be more explicitly mentioned in the Discussion section and we will do so when revising the manuscript.

King, F., Erler, A. R., Frey, S. K., & Fletcher, C. G. (2020). Application of machine learning techniques for regional bias correction of snow water equivalent estimates in Ontario, Canada. *Hydrology and Earth System Sciences*, *24*(10), 4887–4902. https://doi.org/10.5194/hess-24-4887-2020

Steele, H., Small, E. E., & Raleigh, M. S. (2024). Demonstrating a Hybrid Machine Learning Approach for Snow Characteristic Estimation Throughout the Western United States. *Water Resources Research*, *60*(6), e2023WR035805. https://doi.org/10.1029/2023WR035805

L168 – "Nash-Sutcliffe efficiency" I'm not immediately familiar with this metric. Maybe explain briefly?

The NSE is calculated as one minus the ratio of the error variance of the modelled time-series divided by the variance of the observed time-series. It is a commonly known metric in hydrology, so we did not explicitly define it, but we could add it to accommodate for researchers from other domains.

**Feature importance**

To me, it is not expected that downwards shortwave radiation would be the most important feature. You are modeling both the accumulation and ablation season correct? I would expect SW radiation to be very important but only during the ablation season. I'm curious if the feature importances change temporally? This might be interesting insight to include. My guess is that SW radiation has high magnitude SHAP values during the ablation season because $\Delta SWE$ is generally much higher during the ablation than the accumulation season. I'm curious what you would see if you compute relative SHAP values (by normalizing by $\Delta SWE$). I would expect other features (precipitation) to be relatively more important.

Shortwave radiation is indeed most impactful during the ablation period, but it does have some impact on the ML model predictions for the remainder of the year as well.

To test this, we calculated the mean absolute SHAP values for the accumulation and ablation time steps separately (Figure 1), as defined by the sign of the corresponding $\Delta SWE$ prediction. The shortwave radiation is not only the most important feature (on average) during ablation, but also the second most important feature for the accumulation time steps, only below the snowfall rate.

[Figure]

Figure 1: Feature importances of the five top ranking variables, calculated as the mean absolute SHAP value aggregated for all lagged variables, for each ML-based setup and split type. They are ordered according to their average importance for the three ML-based setups. The left subplot shows the results for the accumulation period, that is, for time steps where $\Delta SWE>0$, and the right

subplot for the accumulation, containing the remaining ones. SW radiation refers to the downward shortwave radiation, 'avg' to the daily average, and int to the daily time integral of positive values.

We have also computed the relative mean absolute SHAP values (Table 1), and despite small changes in the feature importance order, the shortwave radiation again features among the most important variables for both split types.

Table 1: Mean relative absolute SHAP values for the top five values according to their average values across the different setups, for both types of split. MSB, PPC and AUG refer to the measurement-based, post-processing and data augmentation setups described in the paper, respectively, and the column mean refers to the average of the three. Regarding the rows, SW radiation refers to the downward shortwave radiation, temp to the temperature, 'avg' to the daily average, and int to the daily time integral of positive values.

Temporal split:

|  | MSB | PPC | AUG | mean |
|---|---|---|---|---|
| Observed SWE | 24.59 | 2.99 | 66.69 | 31.42 |
| SW radiation | 33.23 | 13.08 | 44.53 | 30.28 |
| Air temp. (int) | 20.35 | 12.23 | 13.57 | 15.38 |
| Air temp. (avg) | 10.85 | 12.47 | 14.86 | 12.73 |
| Snowfall rate | 8.57 | 4.30 | 15.56 | 9.48 |

Station split:

|  | MSB | PPC | AUG | mean |
|---|---|---|---|---|
| Observed SWE | 7.86 | 9.60 | 28.30 | 15.25 |
| Air temp. (int) | 10.31 | 13.13 | 14.01 | 12.49 |
| SW radiation | 12.11 | 11.00 | 14.29 | 12.47 |
| Air temp. (avg) | 4.40 | 9.17 | 8.56 | 7.38 |
| Snowfall rate | 4.00 | 6.59 | 10.04 | 6.88 |

A plausible hypothesis is that its importance could be overestimated compared to physical models since it is a good indicator of the seasonality, which the ML model may be using to guide its predictions. Another hypothesis is that at some low-altitude sites like the Col de Porte, that are frequently close to the rain-snow transition in terms of winter temperatures, snowfall and melt may happen in the same day as a result of rapid variations in weather conditions. At such sites, incoming shortwave radiation can hence also modulate the accumulation of SWE (by reducing it at daily scale when there is melt just after) and be therefore a relevant predictor in the accumulation phase.

We will include the above figure and table in the revised manuscript providing a short explanation along the lines of this rebuttal.

L229 – "reaching above a SHAP value unit." What is meant by this?

What was meant is that those variables achieve a mean absolute SHAP value higher than 1. We will re-write that sentence for improved clarity.

Figure 6 – Why did you choose to plot the mean absolute SHAP values? For some features, it may also be interesting to see how the feature impacts the predictions (i.e., increasing or decreasing predicted SWE).

The purpose of this figure was to show the most important variables for SWE prediction and their distinction per hybrid setup and split type, without delving into the more complex relationships that would make the figure less readable. It could even be further compacted by combining the two subplots of that figure into one for easier comparison between spatial and temporal splits, similar to the previous figure on accumulation and ablation, or even replaced by that figure.

For more information regarding the correlation between each predictor and the target, we computed the SHAP violin plots, which show how the values of each variable influence the target. When the predictor goes from blue to red (left to right), it indicates a positive correlation, and from red to blue a negative one. This is most clear for the air temperature, which is red for negative SHAP values and blue for positive ones. Snowfall rate contains very strong positive SHAP values when it is high (meaning it produces a large positive effect to $\Delta SWE$), while its lower values have little influence, as we might expect. These plots will be added to the appendices along with a short discussion of the influence of each variable. This could be even further enriched with scatterplots of specific variables against their SHAP values, as in Figure 7 from the paper.

[Figure]

Figure 2: Violin plots of the SHAP values for the five highest ranking variables in terms of mean absolute value for each type of split and setup. The colour represents the value of the feature from high to low compared to their average. The sign and magnitude of the SHAP values indicate whether the variables have a positive or negative impact on $\Delta SWE$ and how strongly that impact is.

**Discussion**

L281/2 – "The differences in performance concentrate towards the end of the snow period, where the ML-based models particularly improve the timing of the snow melt." Does this indicate that there are substantial errors in the melt dynamics in the physical model?

Our results suggest that there are indeed non-negligible errors in the melt dynamics of Crocus, which the ML models seem to improve. An in-depth analysis of the main causes for that would be highly interesting, although out of scope for our paper. We will add that comment to the Discussion.

**Technical comments**

Mind consistency with 'an ML model' vs. 'a ML model' (for example in the abstract both are used). I personally don't know which is correct but try and be consistent with your usage!

Table 2 – Type "soild" in row 3

L249/250 – Different tenses are used in the same sentence here ('reduced', 'achieves').

L260 – Maybe 'slowest' instead of 'softest' here?

L300 – I'm not sure that 'impoverished' is the best word choice here. Maybe just 'poor' is better.

Thank you for your suggestions, we will incorporate them into the manuscript.

---

## Author Comment (AC2)

**Summary**

Pomarol Moya et al. in "Improving forecasts of snow water equivalent with hybrid machine learning" evaluate various machine learning (ML) based approaches in representing spatiotemporally in sample and out of sample predictions of daily snow water equivalent (SWE) across 10 measurement sites in the Northern Hemisphere, derived from the ESM-SnowMIP project, across 7-20 years. The ML-based estimates are compared to and, in some cases, informed by a physics-based snow model, Crocus. The analysis shows that ML-based models can benefit from learning about daily SWE behavior from both observations and the physics-based model, sometimes helping to offset physics-based snow model errors (e.g., snowmelt rate/snow off date). The order of importance of variables that influence the SWE prediction is also intercompared and rank ordered, many of which relate to snowpack thermodynamics, and could (potentially) be used to inform physics-based model development.

Overall, I think the paper fits within the scope of the Cryosphere and could be, given more work, a valuable contribution. ML-based methods have grown in popularity in recent years, and ML model development/sensitivity analyses like these help to inform where/when ML-based methods are or are not fit for purpose in predicting SWE spatiotemporally. However, I think there are still several major revisions that need to happen prior to this paper being accepted. While I appreciate the authors' thorough analysis, the underlying data (ESM-SnowMIP station network) is quite sparse in space/time and makes me worry about the extensibility of their findings beyond the limited station locations and years assessed. I respect that the authors provided an entire section (Section 4.1) that discusses this very point, but I feel like a more thorough decomposition (e.g., snow climate, elevation, land surface heterogeneity, etc.) of station differences is still needed (given that only a few stations are used to train and assess ML model fidelity). I also think the authors could try and provide more take home messages for the physics-based modeling community from the ML-based results on which variables/processes to target (e.g., fix the long-standing snowmelt rate/snow off date biases in physics-based snow models). Too often it feels like ML-based papers try to show how they can outperform physics-based models rather than how ML-based models/methods can be used to advance physics-based model development. This is particularly salient given that ML-based models poorly predict out of sample in space/time and under different climate scenarios and, therefore, physics-based models appear that they will be needed for the foreseeable future. I have provided, hopefully, constructive comments and suggested edits below for the authors to consider.

Dear reviewer, thank you for your interest in our paper and for your detailed and extensive review. Regarding the main limitations highlighted in this paragraph; we agree that a decomposed analysis given the small size of the dataset could provide more valuable insights, so we will expand on that in our revision of the manuscript. On the other hand, while we hope that our paper is useful for the physics-based community as well, it is important to note that the focus of our paper is to showcase the performance of hybrid models and more generally the benefits of including physical knowledge into ML models, rather than the advancement of purely physics-based modelling. We hope to sufficiently answer all points raised in more detail below.

Comments and suggested edits

Line 24 – cite "The cryosphere has a large impact on the Northern Hemisphere…", maybe with this study…

Huss, M., Bookhagen, B., Huggel, C., Jacobsen, D., Bradley, R.S., Clague, J.J., Vuille, M., Buytaert, W., Cayan, D.R., Greenwood, G., Mark, B.G., Milner, A.M., Weingartner, R. and Winder, M. (2017), Toward mountains without permanent snow and ice. Earth's Future, 5: 418-435. https://doi.org/10.1002/2016EF000514

This paper describes the critical role of the cryosphere in several aspects of the mountain regions, including human livelihood, economy, and ecosystems, and discusses the potential impact of climate change. Thank you for the suggestion; it provides a valuable reference to stress our point, so we will add it.

Line 28 – cite "…due to its spatio-temporal variability…", maybe with this study…

Alonso-González, E., Revuelto, J., Fassnacht, S. R., & López-Moreno, J. I. (2022). Combined influence of maximum accumulation and melt rates on the duration of the seasonal snowpack over temperate mountains. Journal of Hydrology, 608, 127574

This paper discusses the influence of accumulated snow (i.e., peak SWE) and melt rate in snowpack duration for the mountainous areas in the Iberian Peninsula. It does mention the interannual variability of the snowpack and some of its causes, so we will add it. Furthermore, we will expand the literature regarding that claim (Deems, Fassnacht, and Elder 2006; Grünewald et al. 2010).

Deems, Jeffrey S., Steven R. Fassnacht, and Kelly J. Elder. 2006. 'Fractal Distribution of Snow Depth from Lidar Data'. *Journal of Hydrometeorology* 7(2):285–97. doi:10.1175/JHM487.1.

Grünewald, T., M. Schirmer, R. Mott, and M. Lehning. 2010. 'Spatial and Temporal Variability of Snow Depth and Ablation Rates in a Small Mountain Catchment'. *The Cryosphere* 4(2):215–25. doi:10.5194/tc-4-215-2010.

Line 34 – add "machine learning (ML)" as this is the first time it is introduced/defined

Thanks for noticing, it will be added.

Line 36 – "find non-linear structure" – can machine learning only identify non-linear structures or both linear and non-linear?

It refers to both linear and non-linear structures. We will rephrase the sentence by stating that it is not limited to linear ones.

Line 39-40 – you might also include this citation…

Song, Y., W. Tsai, J. Gluck, A. Rhoades, C. Zarzycki, R. McCrary, K. Lawson, and C. Shen, 2024: LSTM-Based Data Integration to Improve Snow Water Equivalent Prediction and Diagnose Error Sources. J. Hydrometeor., 25, 223–237, https://doi.org/10.1175/JHM-D-22-0220.1

This paper implements an LSTM model to predict SWE where lagged observations of either SWE or satellite-observed snow cover fraction are used as predictors. Hence, it is a good addition to the provided literature on that topic and will be included in the next version of the manuscript.

Line 45-46 – this sentence needs a citation for this bold statement. Couldn't the ML models inherent and amplify biases learned from the physics-based models? Also, is there peer-reviewed evidence that ML models can skillfully produce "out of sample" predictions from one mountain/seasonal snow region to another?

There are many examples in the literature that highlight the potential benefits of using hybrid models. For instance, Karpatne et al. (2017) suggest that they may improve consistency with scientific knowledge and produce more generalizable models.

Hybrid models can certainly inherit biases from the physics-based model, so this may introduce some error, but it is precisely because they incorporate observations that bias is mitigated, so long as the observational dataset is representative of the inference domain.

Examples of the ability of hybrid models to extrapolate to untrained locations can be found for hydrological tasks such as streamflow forecasting (e.g., Konapala et al. 2020; Magni et al. 2023), but also for SWE forecasting (Steele et al., 2024).

We will expand the statement along the lines of this answer, adding more references as well.

Karpatne, Anuj, Gowtham Atluri, James H. Faghmous, Michael Steinbach, Arindam Banerjee, Auroop Ganguly, Shashi Shekhar, Nagiza Samatova, and Vipin Kumar. 2017. 'Theory-Guided Data Science: A New Paradigm for Scientific Discovery from Data'. *IEEE Transactions on Knowledge and Data Engineering* 29(10):2318–31. doi:10.1109/TKDE.2017.2720168.

Konapala, Goutam, Shih-Chieh Kao, Scott L. Painter, and Dan Lu. 2020. 'Machine Learning Assisted Hybrid Models Can Improve Streamflow Simulation in Diverse Catchments across the Conterminous US'. *Environmental Research Letters* 15(10):104022. doi:10.1088/1748-9326/aba927.

Magni, Michele, Edwin H. Sutanudjaja, Youchen Shen, and Derek Karssenberg. 2023. 'Global Streamflow Modelling Using Process-Informed Machine Learning'. *Journal of Hydroinformatics* 25(5):1648–66. doi:10.2166/hydro.2023.217.

Steele, Hannah, Eric E. Small, and Mark S. Raleigh. 2024. 'Demonstrating a Hybrid Machine Learning Approach for Snow Characteristic Estimation Throughout the Western United States'. *Water Resources Research* 60(6):e2023WR035805. doi:10.1029/2023WR035805.

Line 60 – change "features" to "conditions"

Good suggestion, we will include it.

Line 71-72 and Line 77-79 – are 10 stations with 7-20 years of measurements enough to properly sample intra- and inter-annual variability of snowpack lifecycles across the Northern Hemisphere? Also, worryingly, only three of the stations are automatic and the others "only [have] manual measurements at irregular intervals". How many snow climates (Sturm and Liston, 2021), elevations, etc. are represented across these stations? Could the authors provide a map plot with automated/manual station lat/lon locations?

Sturm, M., and G. E. Liston, 2021: Revisiting the Global Seasonal Snow Classification: An Updated Dataset for Earth System Applications. J. Hydrometeor., 22, 2917–2938, https://doi.org/10.1175/JHM-D-21-0070.1.

We will expand the description of the station characteristics when describing the data and comment on its representativeness for the Northern Hemisphere in the discussion.

This dataset was compiled for a model intercomparison project, so it does cover a wide range of snowpack conditions. The station locations and characteristics are plotted in Figure 1 below. A vast geographical area across the Northern Hemisphere is covered, although there is some clustering and oversampling in North America. In terms of elevation, only very high-altitude regions (above 4000 m) are missing, despite the automatic stations used for training reaching only up to 2000 m. Finally, most of the snow climates from the provided reference (Sturm and Liston, 2021) are represented, although not all are covered in the manual stations used for testing.

Nevertheless, it is important to note that this is a methodological paper; it seeks to examine the potential of hybrid models, rather than creating a final, model-based SWE product. So, for this purpose, the provided dataset is sufficient. For further discussion into the reasons for choosing this dataset, please refer to the answer to question at L72 from the response to the first reviewer (https://doi.org/10.5194/egusphere-2025-1845-AC1).

[Figure]

*Figure 1: Representation of the a) geographical location, b) elevation and c) snow climate distributions for both automatic and manual stations.*

Line 76 – change "snow water equivalent" to "SWE"

We will change it accordingly.

Line 82-83 – what does it mean that "aggregation methods were performed for some variables according to expert knowledge"? Can you provide readers with the physical basis/intuition for how each of these various aggregation methods for meteorological variables impacts a snowpack's energy/mass balance? This is needed to ensure that the ML method is learning and estimating snow physics for the right reasons.

This sentence aims to convey that variable selection and aggregation methods was not based on data-driven approaches, but rather on their expected influence in snow dynamics according to expert knowledge. We will clarify and expand the aggregation methods in the relevant section. For example, the time integral of positive air temperatures in Celsius is related to the positive degree days, used since decades in the snow modelling community (Hock, 2003).

Hock, Regine. 2003. 'Temperature Index Melt Modelling in Mountain Areas'. Journal of Hydrology 282(1):104–15. doi:10.1016/S0022-1694(03)00257-9.

Line 92 – change "50 layers" to "50 snow layers". Also, I might mention snow temperature or some other thermodynamic variable (given the mention of "energy and mass balance" in the previous sentence(s).

We propose to change that sentence to: "It dynamically adjusts up to 50 snow layers to represent a vertically discretized snow temperature, density and liquid water content profile, and provides a comprehensive evolution of the snow microstructure, thus giving a vision of the snow stratigraphy and its temporal evolution"

Line 99 – change "layer" to "snow layer". Also, does "layer information" mean the dynamic ranges of snow depth when delineating the 50 snow layers over space/time as the snowpack lifecycles evolve?

We will change it accordingly. Layer information means any of the variables reported by Crocus individually for each snow layer rather than for the whole snowpack, in which case we refer to it as "bulk information". We will clarify that in the text.

Line 101 – I would delete "2100 J/kg*K" as it seems like TMI (if an equation to compute cold content is not shown).

The equation is not explicitly written but was described in text, therefore it was deemed relevant to mention the specific heat of ice used for its calculation. However, since multiplying the predictors of a random forest model by a constant factor does not affect its performance, this is indeed not very relevant and will be removed.

Line 103 – why are most of the variables used to train the ML model daily averages?  Was there a sensitivity analysis performed that is not mentioned here?  For example, wouldn't minimum (e.g., nighttime) or maximum (e.g., daytime) temperatures be important too given that the snowpack might refreeze or quickly melt depending on the range of temperatures experienced in a given day?  On Line 114-115 you also mention how there can be a delay in the response of the snowpack (presumably from the erosion of cold content before a phase change occurs) over a 14-day period from the given day.  This also seems to be an argument that some information might be contained in minimum/maximum/etc. of meteorological variables.

As mentioned above in the answer to the Line 82-83 question, the choice of input variables was made based on expert knowledge and no specific sensitivity analysis was performed. The objective was to set the basis of a first model able to capture the most salient features of the SWE dynamics, and there are certainly several avenues of improvement that could be considered; refining the predictors is one of them.

We considered the average to be a good initial approach to aggregate the meteorological variables, but we also performed different aggregations than solely averages. For instance, Tair_int is the daily integral of positive temperatures in Celsius, a well-referenced predictor for snow melt (Hock, 2003), or in its defect, cold content erosion. This led us to discard other aggregation methods which seemed less promising like minimum and maximum, but it does not mean that they might not be useful predictors.

We completely agree that further testing of variable choices and aggregation methods, such as the ones proposed here, remains an interesting research direction for future studies, and we will discuss this limitation in the Discussion section.

Figure 1 – is there a reason that different brackets are used "[]" and "()" to describe time (t)?

Yes, they refer to how the variables were aggregated. We discussed their meaning in the comment regarding Figure 1 in the response to the first reviewer (https://doi.org/10.5194/egusphere-2025-1845-AC1). As we also state there, it certainly requires clarification.

Line 117 – "consecutive daily SWE measurements are available, that is, the automatic stations" does that mean you completely "throw out" seven of the 10 stations data?  If so, I am even more worried about properly sampling intra-annual and inter-annual variability of snowpack lifecycles across the Northern Hemisphere.  1874 days (~5 years) is not very much data to train the ML model on purely observations of SWE/dSWE.  A biggest question, can you more clearly state how the manual measurements are used then?

That is correct; besides the data augmentation (AUG) approach, only the three automatic stations were used for training. The manual measurements were used exclusively for testing purposes in the station split. A diagram explaining the use of manual measurement according to ML model and temporal or station split is provided in Figure 2 in the paper.

Regarding the lack of data, it is important to note that while we only have 1874 samples, those exclude periods without snow, therefore effectively represent much more than five years of data.

Lastly, the success of the AUG strategy, even with limited observational data, highlights its usefulness for forecasting data-scarce variables like SWE. However, we will add that our findings are restricted to stations/climates covered by this dataset in the Discussion.

Line 139 – so you are splitting 1874 days of data into train, validation and test? Are manual measurements used for training, validation, and/or testing too?

Both manual and automatic measurements are considered as ground truth, but the former are only available weekly or less frequently. Because consecutive measurements are required to compute our target (daily ΔSWE), only the automatic measurements can be used for training and hyperparameter tuning. However, because we only need SWE (instead of its daily change) for evaluation purposes, we can still use the manual measurements as a test set in the station split.

Regarding the concern with splitting the already small dataset even further into train, validation and test set; this is exactly what motivated our splitting strategy, which optimizes the available data by smart use of cross validation and re-training after hyperparameter selection. This is explained in Figure 2 in the paper, but we have re-written it below so hopefully it becomes clearer:

The data splits for train, validation and test are different for the temporal and station splits. In the temporal split, 3/5 of the data are used for an initial training of the model and 1/5 as validation for model selection and hyperparameter tuning. Finally, these combined 4/5 of the data are used for a final model training, which is tested in the remaining 1/5. Using cross-validation ensures that our results are robust despite the small test size as we effectively test on the whole available data. In the station split, data from 2 out of the 3 training stations are used for the initial training, and the remaining one for validation of model selection and hyperparameter tuning. This is done so that the hyperparameters are optimized for prediction in untrained stations. Again, this is performed three times in a cross-validation loop to ensure robustness. After selecting the hyperparameters, all of the data from the three automatic stations is used to train the final model, which is tested on the manual samples in the remaining stations.

We will improve the description of the data split types in the manuscript to enhance its intelligibility, as it is still slightly confusing despite our best efforts.

Figure 2 – change "a) the station split and b) the temporal split strategies" to "a) the temporal split and b) station split strategies". Either the a) and b) in the figure is wrong or the caption is wrong.

Yes, the caption is wrong. We will change it accordingly.

Figure 3 – at the moment, a reader (who quickly glances at this plot) might infer that "Sample size auto. stations" of 171, 348, 1355 would mean the number of stations not the number of station measurements used (as I think the authors intend to convey the information). Please change this to be more specific. Also, why would NSE go down for Crocus as more information is used? Is that because model bias becomes more severe as more stations are compared with it?

We agree that the caption it is not 100% clear, we will change it for improved clarity.

Regarding the second point, because Crocus does not use training data, its performance should be independent from the number of evaluation samples in each station. So, the decrease in

performance for stations with larger sample pool for the temporal split is likely coincidental. However, it is not a surprising result that Crocus shows a better SWE simulation at Col de Porte than at other sites, as Col de Porte is historically used for the development of this model and an emphasis has been put all along its development to be able to model the SWE there (Brun et al. 1989, 1992). This finding is already documented in publications (e.g. Menard et al., 2021). Note that Crocus simulations also occasionally helped to detect and correct errors in the meteorological forcing at Col de Porte. It also follows that Crocus may show poorer performance for the simulation of SWE in stations whose characteristics deviate from Col de Porte (medium altitude alpine site with mild winter temperatures, quite wet winters and low snow transport by wind).

We will shortly discuss this in the revised manuscript.

Brun, E., P. David, M. Sudul, and G. Brunot. 1992. 'A Numerical Model to Simulate Snow-Cover Stratigraphy for Operational Avalanche Forecasting'. *Journal of Glaciology* 38(128):13–22. doi:10.3189/S0022143000009552.

Brun, E., E. Martin, V. Simon, C. Gendre, and C. Coleou. 1989. 'An Energy and Mass Model of Snow Cover Suitable for Operational Avalanche Forecasting'. *Journal of Glaciology* 35(121):333–42. doi:10.3189/S0022143000009254.

Menard, C. B., and Coauthors, 2021: Scientific and Human Errors in a Snow Model Intercomparison. *Bull. Amer. Meteor. Soc.*, **102**, E61–E79, https://doi.org/10.1175/BAMS-D-19-0329.1.

Line 184-194 – are these results indicating that Crocus degrades ML model performance in the temporal and enhances ML performance across stations (e.g., comparing AUG result between the two data splits)? Why would this be the case? Also, physically, what does it mean when a model does not perform well in the temporal split but does in the station split?

This statement is true not for hybrid models in general, but only for the AUG setup, which uses Crocus simulations on the stations with manual measurements to artificially increase the number of training samples. When using the post-processing setup (PPC), which uses Crocus predictions only as an additional input, the performance is always better than the "purely" ML approach (MSB), although not by a large margin. The degradation in AUG performance in the temporal split with respect to MSB could be caused by the latter being more specialized (or in ML terms, overfitted) on its three training stations, capturing behaviours specific to them, while the increased number of training stations in AUG results in a better ability to generalize to other stations, but at the cost of station-specific characteristics.

All models performed better in the temporal split than in the station one, meaning that the interannual variability of SWE is much easier to predict than its geographical one. However, this is likely due to the characteristics of our dataset, where relatively long time series are available but only few stations, and no predictors of spatial variation (e.g., topography) were used. More generally, a model not performing well in temporal split may be systematically missing some of the specific processes explaining the snow cover dynamics at a specific station (e.g., snow transport or ablation dynamics due to foehn storms), but it may sufficiently capture most of the generally relevant physical processes of snow and its interaction to the environment so that it performs correctly at station split.

We will further expand the implications of these results in the Discussion section.

Line 196-206 – do they authors know why Crocus systematically underrepresents peak SWE (even when run at a point scale) and melts out the snowpack too early?  Does it have to do with the rain-snow partitioning scheme in the accumulation season?  Could this be enhanced?  For example, Jennings et al. (2018) provides a potential path forward.  Similarly, what might be driving the snowmelt/snow off date bias?  Is there any literature to highlight this as a systematic snow model deficiency?

Jennings, K.S., Winchell, T.S., Livneh, B. et al. Spatial variation of the rain–snow temperature threshold across the Northern Hemisphere. Nat Commun 9, 1148 (2018). https://doi.org/10.1038/s41467-018-03629-7

While an such analysis is out of the scope of this paper, and hence we prefer to leave it out of the manuscript, we hope to add some context in the following response.

The Crocus developers do not have a clear view of the reasons of this underestimation in the ESM-SnowMIP simulations. Actually, the phase partitioning was done by the site-referent researchers (Ménard et al., 2019) and not by the models or the modellers. The methods may be flawed, typically in respect to the findings by Jennings et al., 2018; but this rules out a model-specific bias on this side.

The evaluations of ESM-SnowMIP simulations in Menard et al. 2019, show that this SWE underestimation by Crocus comes with an inhomogeneous bias in albedo (that is typically overestimated at Col de Porte and Swamp Angel; whereas SWE is underestimated at these sites), so that an explanation is hard to find on that side.

Similarly, in these simulations, the SWE underestimation by Crocus comes with a usually underestimated surface temperature (Ménard et al., 2019), that generally should increase the ability of the model to maintain its SWE and is not in line with an anticipated melt.

A behaviour that has been recently highlighted for Crocus (and is not yet published), is that the heat flux from the soil is often erroneous. We observed that at Col de Porte, it can sometimes lead to a complete melt of the first snowfall of the year. This effect could explain part of the systematic SWE negative bias of Crocus, highlighted in Table B1, but does not seem to be involved for the sites WFJ and RME displayed in Fig 4 and 5.
* * *
Ménard, C. B., Essery, R., Barr, A., Bartlett, P., Derry, J., Dumont, M., Fierz, C., Kim, H., Kontu, A., Lejeune, Y., Marks, D., Niwano, M., Raleigh, M., Wang, L., and Wever, N.: Meteorological and evaluation datasets for snow modelling at 10 reference sites: description of in situ and bias-corrected reanalysis data, Earth Syst. Sci. Data, 11, 865–880, https://doi.org/10.5194/essd-11-865-2019, 2019

Line 229-235 – do these differences in meteorological variables/etc. have to do with the stations being located in different snow climates, elevations, shaded/forested regions, etc.?  Do the authors think they have sampled all of these properly in training/testing the ML models?

The importances of meteorological variables are likely dependent on station characteristics, but the current analysis aimed to capture general patterns valid for all locations. As to whether station characteristics are properly sampled, please refer to the question above regarding Line 71-72. In short, the coverage is reasonably good for most station characteristics given the small sample size, which suggests that our results are at least a good indication of what one could expect when

extrapolating to the Northern Hemisphere, but maybe not sufficient to provide strong claims. We will re-write our Results and Discussion to make them more nuanced, stating that these results are a good indication of the variable importances, but more station variety would be needed to provide a definitive answer.

Line 263-264 – do the authors know which stations had more or less sensitivity to lagged meterological variables at +7 day vs 7 day vs 3 day vs 1 day?  Do these stations (and their sensitivities) fall into different snow climates, elevation bands, shaded/forested landscapes, etc.? This sort of information would be important to glean to guide future ML model development/application over a larger spatiotemporal set of stations.

A station-specific analysis of the sensitivity to the lagged meteorological variables would certainly be interesting but falls beyond the scope of this paper. Our current implementation concatenates all stations before computing the importances and would suppose a significant effort to add. We would like to encourage other studies to pursue that question.

Section 3.4.2 – this seems like it should be in the Data and Methods section (or Supplemental Material)

Despite that calling it feature engineering, this section showcases an important result, which is that a version of the post-processing hybrid setup (PPC) which includes additional Crocus variables, and so it has more information available about the snowpack, actually performs similar or worse than the same setup with only the Crocus-reported SWE and ΔSWE. Therefore, we believe it is best to keep it in the results section. Yet, we would like to explicit a bit more the design and purposes of the feature engineering in Material and Methods (specifically sect 2.3.1, line 125), give it a specific name (i.e., PPC-expanded), and keep the analysis of these results in the sect 3.4.2, referring to this set-up name.

Line 276-277 – in Figure 3, didn't the authors show that the AUG model (i.e., hybrid Crocus-ML model) resulted in poorer performance for temporal split (i.e., worse NSE range compared to all physics-based and ML models) and slightly better performance in station split (i.e., NSE range is more constrained and the mean NSE is slightly higher than all physics-based and ML models) than Crocus? Is the difference between Crocus and AUG performance statistically significant/appreciably different for the station split?

The performance of AUG in the temporal split is indeed lower than the other ML approaches, but it is still better than Crocus, which it improves in all test metrics analysed in the study: Nash-Sutcliffe Efficiency (NSE), Root Mean Squared Error (RMSE), Mean Absolute Error (MAE) and Mean Bias (MB).

In the station split, AUG again improves the performance of Crocus (and more so compared to the other ML models) in almost all aspects. It improves the NSE for 6 of the 7 test stations, with an average NSE difference of 0.27 per station, besides improving it also on the entire test dataset from 0.80 to 0.85. Furthermore, it reduces the test RMSE by 12%, and the test MAE by 5%. The only metric in which Crocus achieves better results is the MB, which is almost duplicated (from -34 to -60 mm). So, despite it admittedly being a significantly more biased model, AUG's performance is clearly improved over Crocus.

Line 281-284 – could the tendencies or corrections made by the ML models be used to inform physics-based model development (e.g., to "fix" the snowmelt rate/snow off date bias)? At the very least, could the ML models be used to identify if the variable(s) driving this bias in Crocus (and other physics-based models) are mass or energy related? This could be a major value add from ML models.

This is an interesting idea, and certainly a very desirable direction for future research. However, it does not fully align with the goal of this paper, which is to show the benefits and nuances of using hybrid models. Given our results, it is difficult to assess which variables might be contributing most to biases in Crocus, but we hope more efforts are directed towards that goal in the coming years. We will reflect that by adding this recommendation in the Discussion.

Line 293-295 – what would constitute a "large, representative dataset"? How many days would be needed? How many stations? Etc.

It is hard to define in specific numbers. At the very least, it should cover well the range of the predictors, particularly its edge cases. For example, the limitations of the ML models to correctly predict ΔSWE for high snowfall values (Figure 7 in the paper) indicates that more extreme snowfall events would be needed for improving the ML model training. Similarly, the choice of stations should cover all the different characteristics or locations relevant for snow dynamics (e.g., based on Sturm and Liston, 2021), which is only partially true in our study due to the small sample size. The better performance in the temporal rather than station split indicates that having more locations would be more beneficial than longer time series for our study, but having a good climatic representativity (about 30 years) in the data is also important.

Line 298 – "greater generalization capability" Do you mean Crocus has prognostic, physical equations that can make predictions "out of sample" rather than purely diagnostic/"in sample" inferences (as an ML model arguably does)?

Yes, and we will extend that part in the discussion along these lines.

Line 309 – change "downwards" to "downward"

We will change it accordingly.

Line 313-314 – Do you mean to say something like this "…variable selection should be based on an understanding of the snow climates and geographic heterogeneity (e.g., elevation, forest cover and topographic shading) of the location or region in which the ML model is applied"?

Yes, and we will change the sentence to be more precise following your suggestion.

Line 315-338 – I appreciate that the authors explicitly stated the sample size issue here. I was looking for something like this earlier on though. Maybe a sentence or two in the Methods that references a larger discussion later on in the manuscript?

That would certainly be a great addition, we will include it in the text.

Line 325 – change "specially" to "especially"

We will change it accordingly.

Line 335 – see Song et al. (2024) citation above

It suits the aim of the sentence, so we will add it accordingly.

Line 342 – change "northern hemisphere" to "Northern Hemisphere"

We will change it accordingly.